# On the Role of Neural Collapse in Transfer Learning

**Tomer Galanti**
MIT
galanti@mit.edu

**András György**
DeepMind
agyorgy@deepmind.com

**Marcus Hutter**
DeepMind
mhutter@deepmind.com

## Abstract

We study the ability of foundation models to learn representations for classification that are transferable to new, unseen classes. Recent results in the literature show that representations learned by a single classifier over many classes are competitive on few-shot learning problems with representations learned by special-purpose algorithms designed for such problems. In this paper, we provide an explanation for this behavior based on the recently observed phenomenon that the features learned by overparameterized classification networks show an interesting clustering property, called neural collapse. We demonstrate both theoretically and empirically that neural collapse generalizes to new samples from the training classes, and – more importantly – to new classes as well, allowing foundation models to provide feature maps that work well in transfer learning and, specifically, in the few-shot setting.

## 1 Introduction

In a variety of machine learning applications, we have access to a limited amount of data from the task that we would like to solve, as labeled data is oftentimes scarce and/or expensive. In such scenarios, training directly on the available data is unlikely to produce a hypothesis that generalizes well to new, unseen test samples. A prominent solution to this problem is to apply transfer learning (see, e.g., Caruana, 1995; Bengio, 2012; Yosinski et al., 2014). In transfer learning, we are typically given a large-scale source task (e.g., ImageNet ILSVRC, Russakovsky et al., 2015) and a target task from which we encounter only a limited amount of data. While there are multiple approaches to transfer knowledge between tasks, a popular approach suggests to train a large neural network on a source classification task with a wide range of classes (such as ResNet50, He et al., 2016, MobileNet, Howard et al., 2017 or the VGG network, Simonyan & Zisserman, 2014), and then to train a relatively smaller network (e.g., a linear classifier or a shallow MLP) on top of the penultimate layer of the pretrained network, using the data available in the target task.

Due to the effectiveness of this approach, transfer learning has become a central element in the machine learning toolbox. For instance, using pretrained feature maps is common practice in a variety of applications, including fine-grained classification (Chen et al., 2019a; Huang & Li, 2020; Yang et al., 2018), object detection, (Redmon et al., 2016; Ren et al., 2015; He et al., 2017), semantic segmentation (Long et al., 2015; Chen et al., 2018), or medical imaging (Chen et al., 2019b). In fact, due to the cumulative success of transfer learning, large pretrained models that can be effectively adapted to a wide variety of tasks (Brown et al., 2020; Ramesh et al., 2021) have recently been characterized as foundation models (Bommasani et al., 2021), emphasizing their central role in solving various learning tasks.

Typically, a foundation model is pretrained on a source task at a time when the concrete description of the target task (or target tasks) is not – or only partially – available to the practitioner. Therefore, the ability to precondition or design the training regime of the foundation model to match the target task is limited. As a result, there might be a domain shift between the source and target tasks, such as a different amount of target classes, or a different number of available samples. Hence, intuitively, a foundation model should be fairly generic and applicable in a wide range of problems.

On the other hand, when some specifics of the target tasks are known, often special-purpose algorithms are designed to utilize this information. Such an example is the problem of few-shot learning,

when it is known in advance that the target problems come with a very small training set (Vinyals et al., 2016; Ravi & Larochelle, 2017; Finn et al., 2017; Lee et al., 2019). While these specialized algorithms have significantly improved the state of the art, the recent work of Tian et al. (2020); Dhillon et al. (2020) demonstrated that predictors trained on top of foundation models can also achieve state-of-the-art performance on few-shot learning benchmarks.

Despite the wide range of applications and success of transfer learning and foundation models in particular, only relatively little is known theoretically why transfer learning is possible between two tasks in the setting mentioned above.

**Contributions.** In this paper we present a new perspective on transfer learning with foundation models, based on the phenomenon of neural collapse (Papyan et al., 2020). Informally, neural collapse identifies training dynamics of deep networks for standard classification tasks, where the features (the output of the penultimate layer) associated with training samples belonging to the same class concentrate around their class feature mean. We demonstrate that this property generalizes to new data points and new classes (e.g., the target classes), when the model is trained on a large set of classes with many samples per class. In addition, we show that in the presence of neural collapse, training a linear classifier on top of the learned penultimate layer can be done using only few samples. We verify these findings both theoretically and via experiments. In particular, our results provide a compelling explanation for the empirical success of foundation models, as observed, e.g., by Tian et al. (2020); Dhillon et al. (2020). In Appendix B.2, we show evidence for neural collapse for feature maps obtained from lower layers of the network, as well, although to a lesser extent.

The rest of the paper is organized as follows: Additional related work is discussed in Section 1.1. The problem setting, in particular, foundation models, are introduced in Section 2. Neural collapse is described in Section 3. Our theoretical analysis is presented in Section 4: generalization to unseen examples is considered in Section 4.1, to new classes in Section 4.2, while the effect of neural collapse on the classification performance is discussed in Section 4.3. Experiments are presented in Section 5, and conclusions are drawn in Section 6. Proofs and additional experiments are relegated to the appendix.

## 1.1 OTHER RELATED WORK

Various publications, such as Baxter (2000); Maurer et al. (2016); Pentina & Lampert (2014); Galanti et al. (2016); Khodak et al. (2019); Du et al. (2021), suggested different frameworks to theoretically study various multi-task learning settings. However, all of these works consider learning settings in which the learning algorithm is provided with a sequence of learning problems, and therefore, are unable to characterize the case where only one source task is available.

In contrast, we propose a theoretical framework in which the learning algorithm is provided with one classification task and is required to learn a representation of the data that can be adapted to solve new unseen tasks using few samples. This kind of modeling is aligned with the common practice of transfer learning using deep neural networks.

While in unsupervised domain adaptation (DA) (Ben-David et al., 2006; Mansour et al., 2009) one typically has access to a single source and a single target task, one does not have access to labels from the former, making the problem ill-posed by nature and shaping the algorithms to behave differently than those that are provided with labeled samples from the target task. In particular, although in transfer learning we typically train the feature map on the source dataset, independently of the target task, in DA this would significantly limit the algorithm's ability to solve the target task.

## 2 PROBLEM SETUP

We consider the problem of training foundation models, that is, general feature representations which are useful on a wide range of learning tasks, and are trained on some auxiliary task. To model this problem, we assume that the final target task we want to solve is a $k$-class classification problem $T$ (the *target* problem), coming from an unknown distribution $\mathcal{D}$ over such problems, and the auxiliary task where the feature representation is learned on an $l$-class classification problem, called the *source* problem. Formally, the target task is defined by a distribution $P$ over samples $(x, y) \in \mathcal{X} \times \mathcal{Y}_k$, where $\mathcal{X} \subset \mathbb{R}^d$ is the instance space, and $\mathcal{Y}_k$ is a label space with cardinality $k$. To simplify the presentation, we use one-hot encoding for the label space, that is, the labels are represented by the unit vectors in $\mathbb{R}^k$, and $\mathcal{Y}_k = \{e_i : i = 1, \ldots, k\}$ where $e_i \in \mathbb{R}^k$ is the $i$th standard

unit vector in $\mathbb{R}^k$; with a slight abuse of notation, sometimes we will also write $y = i$ instead of $y = e_i$. For a pair $(x, y)$ with distribution $P$, we denote by $P_i$ the class conditional distribution of $x$ given $y = i$ (i.e., $P_i(\cdot) = \mathbb{P}[x \in \cdot \mid y = i]$).

A classifier $h : \mathcal{X} \to \mathbb{R}^k$ assigns a *soft* label to an input point $x \in \mathcal{X}$, and its performance on the target task $T$ is measured by the risk

$$L_T(h) = \mathbb{E}_{(x,y) \sim P}[\ell(h(x), y)] , \tag{1}$$

where $\ell : \mathbb{R}^k \times \mathcal{Y}_k \to [0, \infty)$ is a non-negative loss function (e.g., zero-one or cross-entropy losses).

Our goal is to learn a classifier $h$ from some training data $S = \{(x_i, y_i)\}_{i=1}^n$ of $n$ independent and identically distributed (i.i.d.) samples drawn from $P$. However, when $n$ is small and the classification problem is complicated, this might not be an easy problem. To facilitate finding a good solution, we aim to find a classifier of the form $h = g \circ f$, where $f : \mathbb{R}^d \to \mathbb{R}^p$ is a feature map from a family of functions $\mathcal{F} \subset \{f' : \mathbb{R}^d \to \mathbb{R}^p\}$ and $g \in \mathcal{G} = \{g' : \mathbb{R}^p \to \mathbb{R}^k\}$ is a classifier used on the feature space $\mathbb{R}^p$. The idea is that the feature map $f$ is learned on some other problem where more data is available, and then $g$ is trained to solve the hopefully simpler classification problem of finding $y$ based on $f(x)$, instead of $x$. That is, $g$ is actually a function of the modified training data $\{(f(x_i), y_i)\}_{i=1}^n$; to emphasize this dependence, we denote the trained classifier by $g_{f,S}$.

We assume that the auxiliary (source) task helping to find $f$ is an $l$-class classification problem over the same sample space $\mathcal{X}$, given by a distribution $\tilde{P}$, and here we are interested in finding a classifier $\tilde{h} : \mathcal{X} \to \mathbb{R}^l$ of the form $\tilde{h} = \tilde{g} \circ f$, where $\tilde{g} \in \tilde{\mathcal{G}} \subset \{g' : \mathbb{R}^p \to \mathbb{R}^l\}$ is a classifier over the feature space $f(\mathcal{X}) = \{f(x) : x \in \mathcal{X}\}$. Given a training dataset $\tilde{S} = \{(\tilde{x}_i, \tilde{y}_i)\}_{i=1}^m$, both components of the trained classifier, denoted by $f_{\tilde{S}}$ and $\tilde{g}_{\tilde{S}}$ are trained on $\tilde{S}$, with the goal of minimizing the risk in the source task, given by

$$L_S(\tilde{h}_{\tilde{S}}) = \mathbb{E}_{(x,y) \sim \tilde{P}}[\ell(\tilde{g}_{\tilde{S}}(f_{\tilde{S}}(x)), y)]$$

(note that with a slight abuse of notation we used the same loss function as in (1), although they operate over spaces of different dimensions). A standard assumption is that $\tilde{S}$ is drawn i.i.d. form $\tilde{P}$; however, in this work instead we will use hierarchical schemes where first classes are sampled according to their marginals in $\tilde{P}$, and then for any selected class $c$, training samples $\tilde{S}_c$ are chosen from the corresponding class-conditional distribution (or just class conditional, for short) $\tilde{P}_c$; the specific assumptions on the distribution of $\tilde{S}$ will be given when needed. Since $f, \tilde{g}$ and $\tilde{h}$ always depend on the source data only, to simplify the notation we will often drop the $\tilde{S}$ subscript whenever this does not cause any confusion.

In a typical setting $\tilde{h}$ is a deep neural network, $f$ is the representation in the last internal layer of the network (i.e., the penultimate *embedding* layer), and $\tilde{g}$, the last layer of the network, is a linear map; similarly $g$ in the target problem is often taken to be linear. The learned feature map $f$ is called a *foundation model* (Bommasani et al., 2021) when it can be effectively used in a wide range of tasks. In particular, its effectiveness can be measure by its expected performance over target tasks:

$$L_{\mathcal{D}}(f) = \mathbb{E}_{T \sim \mathcal{D}} \mathbb{E}_{S \sim P^n}[L_T(g_S \circ f)]$$

(recall that $P$ is the distribution associated with task $T$).

Notice that while the feature map $f$ is evaluated on the distribution of target tasks determined by $\mathcal{D}$, the training of $f$ in a foundation model, as described above, is fully agnostic of this target. In contrast, several transfer learning methods (such as few-shot learning algorithms) have been developed which optimize $f$ not only as a function of its training data $\tilde{S}$, but also based on some properties of $\mathcal{D}$, such as the number of classes and the number of samples per class in $S$.

Perhaps surprisingly, recent studies (Tian et al., 2020; Dhillon et al., 2020) demonstrate that the target-agnostic training of foundation models (or some slight variation of them, such as transductive learning) are competitve with such special purpose algorithm. In the rest of the paper we analyze this phenomenon, and provide an explanation through the recent concept of neural collapse.

**Notation.** For an integer $k \geq 1$, $[k] = \{1, \ldots, k\}$. $\|\cdot\|$ stands for the Euclidean norm. For a set $A = \{a_i\}_{i=1}^n \subset B$ and a function $u \in \mathcal{U} \subset \{u' : B \to \mathbb{R}\}$, we define $u(A) = \{u(a_1), \ldots, u(a_n)\}$ and $\mathcal{U}(A) = \{u(A) : u \in \mathcal{U}\}$. For a distribution $Q$ over $\mathcal{X} \subset \mathbb{R}^d$ and $u : \mathcal{X} \to \mathbb{R}^p$, we denote by

$\mu_u(Q) = \mathbb{E}_{x \sim Q}[u(x)]$ and by $\mathrm{Var}_u(Q) = \mathbb{E}_{x \sim Q}[\|u(x) - \mu_u(Q)\|^2]$ the mean and variance of $u(x)$ for $x \sim Q$. For $A = \{a_i\}_{i=1}^n \subset \mathbb{R}$, we denote $\mathrm{Avg}_{i=1}^n[a_i] = \frac{1}{n}\sum_{i=1}^n a_i$. For a finite set $A$, we denote by $U[A]$ the uniform distribution over $A$.

## 3 NEURAL COLLAPSE

Neural collapse (NC) is a recently discovered phenomenon in deep learning (Papyan et al., 2020; Han et al., 2021): it has been observed that during the training of deep networks for standard classification tasks, the features (the output of the penultimate, a.k.a. embedding layer) associated with training samples belonging to the same class concentrate around the mean feature value for the same class. More concretely, Papyan et al. (2020) observed essentially that the ratio of the within-class variances and the distances between the class means converge to zero. They also noticed that asymptotically the class-means (centered at their global mean) are not only linearly separable, but are actually maximally distant and located on a sphere centered at the origin up to scaling (they form a simplex equiangular tight frame), and furthermore, that the behavior of the last-layer classifier (operating on the features) converges to that of the nearest-class-mean decision rule.

On the theoretical side, already Papyan et al. (2020) showed the emergence of neural collapse for a Gaussian mixture model and linear score-based classifiers. Poggio & Liao (2020b;a); Mixon et al. (2020) theoretically investigated whether neural collapse occurs when training neural networks under MSE-loss minimization with different settings of regularization. Furthermore, Zhu et al. (2021) showed that in certain cases, any global optimum of the classical cross-entropy loss with weight decay in the unconstrained features model satisfies neural collapse.

Papyan et al. (2020) defined neural collapse via identifying that the within-class variance normalized by the intra-class-covariance tends to zero towards the end of the training (see Appendix B.4 for details). In this paper, we work with a slightly different variation of within-class variation collapse, which will be later connected to the clusterability of the sample feature vectors. For a feature map $f$ and two (class-conditional) distributions $Q_1, Q_2$ over $\mathcal{X}$, we define their *class-distance normalized variance* (CDNV) to be

$$V_f(Q_1, Q_2) = \frac{\mathrm{Var}_f(Q_1) + \mathrm{Var}_f(Q_2)}{2\|\mu_f(Q_1) - \mu_f(Q_2)\|^2} .$$

For finite sets $S_1, S_2 \subset \mathcal{X}$ we define $V_f(S_1, S_2) = V_f(U[S_1], U[S_2])$. Our version of *neural collapse* (at training) asserts that $\lim_{t \to \infty} \mathrm{Avg}_{i \neq j \in [l]}[V_{f_t}(\tilde{S}_i, \tilde{S}_j)] = 0$. Intuitively, when decreasing the variance of the features of a given class in comparison to its distance from another class, we expect to be able to classify the features into the classes with a higher accuracy. This definition is essentially the same as that of Papyan et al. (2020), making their empirical observations about neural collapse valid for our definition (as also demonstrated in our experiments), but our new definition simplifies the theoretical analysis. Furthermore, the theoretical results of Zhu et al. (2021); Mixon et al. (2020); Poggio & Liao (2020a;b) also identify settings in which the feature map $f$ induced by critical points of the training loss satisfies $\mathrm{Avg}_{i \neq j} V_f(\tilde{S}_i, \tilde{S}_j) = 0$ (for large enough sample sizes).

## 4 NEURAL COLLAPSE ON UNSEEN DATA

In this section we theoretically analyze neural collapse in the setting of Section 2, with particular attention of variance collapse on data not used in the training of the classifier. First, in Section 4.1 we show that if neural collapse happens on the training set $\tilde{S}$, then it generalizes to unseen samples from the same task under mild, natural assumptions (given the sample size is large enough). Next we show, in Section 4.2, that one can also expect neural collapse to happen over new classes when the classes in the source and target tasks are selected randomly from the same class distributions (given many source classes). The significance of these results is that they show that the feature representations learned during training are immediately useful in other tasks; we show in Section 4.3 how this leads to low classification error in few-shot learning.

### 4.1 GENERALIZATION TO NEW SAMPLES FROM THE SAME CLASS

In this section, we provide a generalization bound on the CDNV, $V_f(\tilde{P}_i, \tilde{P}_j)$, between two source class-conditional distributions $\tilde{P}_i$ and $\tilde{P}_j$ in terms of its empirical counterpart, $V_f(\tilde{S}_i, \tilde{S}_j)$, and cer-

tain generalization gap terms bounding the difference between expectations and empirical averages for $f$ and its variants, where $f$ is the output of the learning algorithm with access to $\tilde{S}$. Assume that for all $c \in [l]$, $\tilde{S}_c$ is a set of $m_c$ i.i.d. samples drawn from $\tilde{P}_c$, and for any $\delta \in (0,1)$, let $\epsilon_1(\tilde{P}_c, m_c, \delta)$ and $\epsilon_2(\tilde{P}_c, m_c, \delta)$ be the smallest positive values such that with probability at least $1-\delta$, the learning algorithm returns a function $f \in \mathcal{F}$ that satisfies

$$\left\| \mathbb{E}_{x \sim \tilde{P}_c}[f(x)] - \mathrm{Avg}_{x \in \tilde{S}_c}[f(x)] \right\| \leq \epsilon_1(\tilde{P}_c, m_c, \delta) =: \epsilon_1^c(\delta);$$

$$\left| \mathbb{E}_{x \sim \tilde{P}_c}[\|f(x)\|^2] - \mathrm{Avg}_{x \in \tilde{S}_c}[\|f(x)\|^2] \right| \leq \epsilon_2(\tilde{P}_c, m_c, \delta) =: \epsilon_2^c(\delta).$$

Typically, these quantities can be upper bounded using Rademacher complexities (Bartlett & Mendelson, 2002) related to $\mathcal{F}$, scaling usually as $\mathcal{O}(\sqrt{\log(1/\delta)/m_c})$ (for a fixed $c$), as we show in Proposition 3 (in Appendix E) for ReLU neural networks with bounded weights. Next, we present our bound on the CDNV of the source distributions; the proof is relegated to Appendix C.

**Proposition 1.** *Fix two source classes, $i$ and $j$ with distributions $\tilde{P}_i$ and $\tilde{P}_j$, and let $\delta \in (0,1)$. Let $\tilde{S}_c \sim \tilde{P}_c^{m_c}$ for $c \in \{i,j\}$. Let*

$$A = \frac{\epsilon_1^i(\delta/4) + \epsilon_1^j(\delta/4)}{\|\mu_f(\tilde{P}_i) - \mu_f(\tilde{P}_j)\|} \quad and \quad B = \frac{\mathrm{Avg}_{c \in \{i,j\}}\left[ \epsilon_2^c(\delta/4) + 2\|\mu_f(\tilde{P}_c)\| \cdot \epsilon_1^c(\delta/4) + \epsilon_1^c(\delta/4)^2 \right]}{\|\mu_f(\tilde{S}_i) - \mu_f(\tilde{S}_j)\|^2}.$$

*Then, with probability at least $1 - \delta$ over $\tilde{S}$, we have $V_f(\tilde{P}_i, \tilde{P}_j) \leq \left(V_f(\tilde{S}_i, \tilde{S}_j) + B\right)(1 + A)^2$.*

Writing out the bracket, the bound in the proposition has several terms. The first one is the empirical CDNV $V_f(\tilde{S}_i, \tilde{S}_j)$ between the two datasets $\tilde{S}_i \sim \tilde{P}_i^{m_i}$ and $\tilde{S}_j \sim \tilde{P}_j^{m_j}$. This term is assumed to be small by the neural collapse phenomenon during training. The rest of the terms are proportional to the generalization gaps $\epsilon_1^c(\delta/4)$ and $\epsilon_2^c(\delta/4)$ — as discussed above, typically we expect these terms to scale as $\mathcal{O}(\sqrt{\log(1/\delta)/m_c})$. In addition, according to the theoretical analyses of neural collapse available in the literature (Papyan et al., 2020; Mixon et al., 2020; Zhu et al., 2021; Rangamani et al., 2021), under certain conditions the function $f$ converges to a solution for which $\{\mu_f(\tilde{S}_i)\}_{i=1}^l$ form a simplex equiangular tight frame (ETF), that is, after centering their global mean, $\mu_f(\tilde{S}_i)$ are of equal length, and $\|\mu_f(\tilde{S}_i) - \mu_f(\tilde{S}_j)\|$ are also equal and maximized for all $i \neq j$. This implies that if $f$ is properly normalized, the distance $\|\mu_f(\tilde{S}_i) - \mu_f(\tilde{S}_j)\|$ is lower bounded by a constant, and hence $A$ and $B$ indeed are small (in Appendix B.3 we demonstrate empirically that the minimum distances of the class means are indeed not too small in the scenarios we consider). Finally, we also note that with probability $1 - \delta$, we have $\|\mu_f(\tilde{P}_i) - \mu_f(\tilde{P}_j)\| \geq \|\mu_f(\tilde{S}_i) - \mu_f(\tilde{S}_j)\| - \epsilon_1^i(\delta/2) - \epsilon_1^j(\delta/2)$.

Hence, if $m_i$ and $m_j$ are large enough, $\|\mu_f(\tilde{P}_i) - \mu_f(\tilde{P}_j)\|$ is larger than a constant with high probability. Therefore, assuming $m_i, m_j$ are large, if $V_f(\tilde{S}_i, \tilde{S}_j)$ is small, then we expect $V_f(\tilde{P}_i, \tilde{P}_j)$ to also be small. That is, if neural collapse emerges in the training data of two source classes, we should also expect it to emerge in unseen samples of the same classes.

## 4.2 Neural Collapse Generalizes to New Classes

Previously, we showed that if the CDNV is minimized by the learning algorithm on the training data, we expect it to be small for unseen source samples. As a next step, we show that if neural collapse emerges in the set of source classes, we can also expect it to emerge in new, unseen target classes.

To analyze this scenario, we essentially treat class-conditional distributions as data points on which the feature map $f$ is trained (in a noisy manner, depending on the actual samples), and apply standard techniques to derive generalization bounds to new data points, which, in this case, are class-conditional distributions. Accordingly, we assume that both the source and the target classes come from a distribution over $\mathcal{D}_{\mathcal{C}}$ over a set of classes $\mathcal{C}$. Each class is represented by its class-conditional distribution, and hence, with a slight abuse of notation, we say that the source class-conditional distributions $\tilde{\mathcal{P}} = \{\tilde{P}_i\}_{i=1}^l$ are selected according to $\mathcal{D}_{\mathcal{C}}(\tilde{P}_1, \dots, \tilde{P}_l \mid \tilde{P}_i \neq \tilde{P}_j \text{ for all } i \neq j \in [l])$. Below we show that if neural collapse emerges in the source classes, it also emerges in unseen target classes $c \neq c'$ with class-conditional distributions $P_c, P_{c'} \sim \mathcal{D}_{\mathcal{C}}(P_c, P_{c'} \mid P_c \neq P_{c'})$ in the sense that $V_f(P_c, P_{c'})$ is expected to be small. Our bound below applies a uniform convergence argument based on Rademacher complexities. The Rademacher complexity of a set $Y \subset \mathbb{R}^n$ is defined as

$R(Y) = \mathbb{E}_\epsilon \sup_{y \in Y} [\langle \epsilon, y \rangle]$, where $\epsilon = (\epsilon_1, \ldots, \epsilon_n) \sim U[\{\pm 1\}^n]$. For a given feature map $f \in \mathcal{F}$, we also define a mapping $H_f : \mathcal{C} \to \mathbb{R}^{p+1}$ as $H_f(P_c) = (\mu_f(P_c), \mathrm{Var}_f(P_c))$, and for a class of functions $\mathcal{F}^*$, we define $H_{\mathcal{F}^*}(\tilde{\mathcal{P}}) = \{(H_f(\tilde{P}_c))_{c=1}^l : f \in \mathcal{F}^*\}$.

**Proposition 2.** *Let $\mathcal{F}^* \subset \mathcal{F}$ be any finite set of functions with $\Delta(\mathcal{F}^*) = \inf_{f \in \mathcal{F}^*} \inf_{P_c \neq P_{c'}} \|\mu_f(P_c) - \mu_f(P_{c'})\| > 0$. Then, with probability at least $1 - \delta$ over the selection of source class distributions $\tilde{\mathcal{P}}$,*

$$\mathbb{E}_{P_c \neq P_{c'}}[V_f(P_c, P_{c'})] \leq \mathrm{Avg}_{i \neq j}\Big[V_f(\tilde{P}_i, \tilde{P}_j)\Big] + \left(8 + \frac{16 \sup\limits_{f \in \mathcal{F}^*, P' \in \mathcal{C}} \mathrm{Var}_f(P')}{\Delta(\mathcal{F}^*)}\right) \cdot \frac{\sqrt{2\pi \log(l)}\mathbb{E}[R(H_{\mathcal{F}^*}(\tilde{\mathcal{P}}))]}{(l-1) \cdot \Delta(\mathcal{F}^*)^2}$$

$$+ \left(1 + \frac{4 \sup_{x \in \mathcal{X}, f \in \mathcal{F}^*} \|f(x)\|}{\Delta(\mathcal{F}^*)}\right) \cdot \frac{2\sqrt{\log(1/\delta)} \cdot \sup\limits_{f \in \mathcal{F}^*, P' \in \mathcal{C}} \mathrm{Var}_f(P')}{\sqrt{l} \cdot \Delta(\mathcal{F}^*)^2} \ .$$

The above proposition, proved in Appendix D (based on Corollary 3 of Maurer & Pontil, 2019), provides an upper bound on the discrepancy between the expected value of $\mathbb{E}_{P_c \neq P_{c'}}[V_f(P_c, P_{c'})]$ and the averaged value $\mathrm{Avg}_{i \neq j}[V_f(\tilde{P}_i, \tilde{P}_j)]$ across a set of source distributions $\tilde{P}_1, \ldots, \tilde{P}_l$ that were sampled from $\mathcal{D}_{\mathcal{C}}$. In addition, we treat $\mathcal{F}^*$ as a set of candidate functions from which the learning algorithm selects its candidates. Similarly to Corollary 3 of Maurer & Pontil (2019), this set is assumed to be finite only for technical (measurability) reasons.

The proposition shows that if $\mathrm{Avg}_{i \neq j}[V_f(\tilde{P}_i, \tilde{P}_j)]$ is small, we expect $\mathbb{E}_{P_c \neq P_{c'}}[V_f(P_c, P_{c'})]$ to be small, if the source data is representative enough, as discussed below. First note that by Proposition 1, each term $V_f(\tilde{P}_i, \tilde{P}_j)$ is expected to be small in the presence of neural collapse (for large enough training sets $\tilde{S}_c$), so their average $\mathrm{Avg}_{i \neq j}[V_f(\tilde{P}_i, \tilde{P}_j)]$ is also small. According to Proposition 4 in Appendix E, the Rademacher complexity in the bound scales as $\mathcal{O}(p\sqrt{l})$ for ReLU networks with bounded weights, so the corresponding term in the bound is $\mathcal{O}(p/(\sqrt{l}\Delta(\mathcal{F}^*)^2))$, which is of similar order as the last term (neglecting the supremum terms and the dependence on $p$ for now).

Hence, the bound tends to zero as the number of source classes $l$ increases as long as $\Delta(\mathcal{F}^*)$ is not too small (in particular, is not zero). While there is no way to guarantee this (e.g., if two class-conditional distributions are too close, $\Delta(\mathcal{F}^*)$ can be very small), we can say that if the feature maps $f \in \mathcal{F}^*$ keep the classes apart, the bound may become reasonably small. If the feature dimension $p$ is large and $f \in \mathcal{F}^*$ are similar to random maps, this assumption typically holds (e.g., if $f$ is not a constant function, which can be expected with a proper training method). Even when the number of classes is very large, the bound can be reasonable: if the corresponding feature means are more-or-less uniformly distributed in a $p$-dimensional unit cube for all $f \in \mathcal{F}^*$, $\Delta(\mathcal{F}^*) = \Omega(\sqrt{p} \cdot |\mathcal{C}|^{-2/p})$ (see Lemma 2 in Appendix F), and hence combining with the dependence of the Rademacher complexity on $l$ and $p$, and the fact that $\mathrm{Var}_f(P') \leq p$ and $\sup_{x,f} \|f(x)\| \leq \sqrt{p}$, the bound is $\mathcal{O}(\sqrt{p/l} \cdot |\mathcal{C}|^{6/p})$ which tends to zero as $l$ increases.

### 4.3 CDNV AND CLASSIFICATION ERROR

Next, we study the relation between the CDNV and the classification performance it induces. Consider a balanced $k$-class classification problem with class distributions $P_1, \ldots, P_k$, datasets $S_c \sim P_c^{n_c}$, $c \in [k]$ and $n_1 = \ldots = n_k$. Let $S = \cup_{c \in [k]} S_c$ denote all the samples and consider the 'nearest empirical mean' classifier $h_{S,f}(x) = \arg\min_{c \in [k]} \|f(x) - \mu_f(S_c)\|$ (note that $h_{S,f}$ is a classifier with convex polytope cells on top of $f$, and it is a linear classifier when $k = 2$). Then, as it is shown in Proposition 5 in Appendix G the expected classification error of $h_{S,f}$ is upper bounded by $16(k-1)\mathrm{Avg}_{i \neq j} V_f(P_i, P_j)(1 + 1/n_c)$ and if $f \circ P_1$ and $f \circ P_2$ are spherically symmetric, then the classification error is upper bounded by $16(k-1)\mathrm{Avg}_{i \neq j} V_f(P_i, P_j)(1/p + 1/n_c)$. Therefore, in case of neural collapse (i.e., small $\mathrm{Avg}_{i \neq j} V_f(P_i, P_j)$), the error probability is small, explaining the success of foundation models in the low-data regime (such as in few-shot learning problems) in the presence of neural collapse. Putting together Propositions 1 and 2, and Proposition 5, it follows that, with high probability (over the selection of the source, as well as the source samples), the expected error (over the target classes and target samples) of a nearest class-mean classifier over the target classes can be bounded by the average neural collapse over the source training samples plus

| Method | Architecture | Mini-ImageNet | | CIFAR-FS | | FC-100 | |
|---|---|---|---|---|---|---|---|
| | | 1-shot | 5-shot | 1-shot | 5-shot | 1-shot | 5-shot |
| Matching Networks (Vinyals et al., 2016) | 64-64-64-64 | $43.56 \pm 0.84$ | $55.31 \pm 0.73$ | - | - | - | - |
| LSTM Meta-Learner (Ravi & Larochelle, 2017) | 64-64-64-64 | $43.44 \pm 0.77$ | $60.60 \pm 0.71$ | - | - | - | - |
| MAML (Finn et al., 2017) | 32-32-32-32 | $48.70 \pm 1.84$ | $63.11 \pm 0.92$ | $58.9 \pm 1.9$ | $71.5 \pm 1.0$ | - | - |
| Prototypical Networks (Snell et al., 2017) | 64-64-64-64 | $49.42 \pm 0.78^\dagger$ | $68.20 \pm 0.66^\dagger$ | $55.5 \pm 0.7$ | $72.0 \pm 0.6$ | $35.3 \pm 0.6$ | $48.6 \pm 0.6$ |
| Relation Networks (Sung et al., 2018) | 64-96-128-256 | $50.44 \pm 0.82$ | $65.32 \pm 0.7$ | $55.0 \pm 1.0$ | $69.3 \pm 0.8$ | - | - |
| SNAIL (Mishra et al., 2018) | ResNet-12 | $55.71 \pm 0.99$ | $68.88 \pm 0.92$ | - | - | - | - |
| TADAM (Oreshkin et al., 2018) | ResNet-12 | $58.50 \pm 0.30$ | $76.7 \pm 0.3$ | - | - | $40.1 \pm 0.4$ | $56.1 \pm 0.4$ |
| AdaResNet (Munkhdalai et al., 2018) | ResNet-12 | $56.88 \pm 0.62$ | $71.94 \pm 0.57$ | - | - | - | - |
| Dynamics Few-Shot (Gidaris & Komodakis, 2018) | 64-64-128-128 | $56.20 \pm 0.86$ | $73.0 \pm 0.64$ | - | - | - | - |
| Activation to Parameter (Qiao et al., 2018) | WRN-28-10 | $59.60 \pm 0.41^\dagger$ | $73.74 \pm 0.19^\dagger$ | - | - | - | - |
| R2D2 (Bertinetto et al., 2019) | 96-192-384-512 | $51.2 \pm 0.6$ | $68.8 \pm 0.1$ | $65.3 \pm 0.2$ | $79.4 \pm 0.1$ | - | - |
| Shot-Free (Ravichandran et al., 2019) | ResNet-12 | $59.04 \pm n/a$ | $77.64 \pm n/a$ | $69.2 \pm n/a$ | $84.7 \pm n/a$ | - | - |
| TEWAM Qiao et al. (2019) | ResNet-12 | $60.07 \pm n/a$ | $75.90 \pm n/a$ | $70.4 \pm n/a$ | $81.3 \pm n/a$ | - | - |
| TPN (Liu et al., 2019) | ResNet-12 | $55.51 \pm 0.86$ | $75.64 \pm n/a$ | - | - | - | - |
| LEO (Rusu et al., 2019) | WRN-28-10 | $61.76 \pm 0.08^\dagger$ | $77.59 \pm 0.12^\dagger$ | - | - | - | - |
| MTL (Sun et al., 2019) | ResNet-12 | $61.20 \pm 1.80$ | $75.50 \pm 0.80$ | - | - | - | - |
| OptNet-RR (Lee et al., 2019) | ResNet-12 | $61.41 \pm 0.61$ | $77.88 \pm 0.46$ | $72.6 \pm 0.7$ | $84.3 \pm 0.5$ | $40.5 \pm 0.6$ | $57.6 \pm 0.9$ |
| MetaOptNet (Lee et al., 2019) | ResNet-12 | $62.64 \pm 0.61$ | $78.63 \pm 0.46$ | $72.0 \pm 0.7$ | $84.2 \pm 0.5$ | $41.1 \pm 0.6$ | $55.3 \pm 0.6$ |
| Transductive Fine-Tuning (Dhillon et al., 2020) | WRN-28-10 | $65.73 \pm 0.68$ | $78.40 \pm 0.52$ | $76.58 \pm 0.68$ | $85.79 \pm 0.5$ | $43.16 \pm 0.59$ | $57.57 \pm 0.55$ |
| Distill-simple (Tian et al., 2020) | ResNet-12 | $62.02 \pm 0.63$ | $79.64 \pm 0.44$ | $71.5 \pm 0.8$ | $86.0 \pm 0.5$ | $42.6 \pm 0.7$ | $59.1 \pm 0.6$ |
| Distill (Tian et al., 2020) | ResNet-12 | $64.82 \pm 0.60$ | $82.14 \pm 0.43$ | $73.9 \pm 0.8$ | $86.9 \pm 0.5$ | $44.6 \pm 0.7$ | $60.9 \pm 0.6$ |
| Ours (simple) | WRN-28-4 | $58.12 \pm 1.19$ | $72.0 \pm 0.99$ | $68.81 \pm 1.20$ | $81.49 \pm 0.98$ | $44.96 \pm 1.14$ | $57.21 \pm 10.89$ |
| Ours (lr scheduling) | WRN-28-4 | $60.37 \pm 1.25$ | $72.35 \pm 0.99$ | $70.0 \pm 1.29$ | $81.39 \pm 0.96$ | $43.42 \pm 1.0$ | $54.14 \pm 1.1$ |
| Ours (lr scheduling + model selection) | WRN-28-4 | $61.27 \pm 1.14$ | $74.74 \pm 0.76$ | $72.37 \pm 1.12$ | $82.94 \pm 0.89$ | $45.81 \pm 1.27$ | $56.85 \pm 1.30$ |

Table 1: **Comparison to prior work on Mini-ImageNet, CIFAR-FS, and FC-100 on 1-shot and 5-shot 5-class classification.** Reported numbers are test target classification accuracy of various methods for various data sets. The notation a-b-c-d denotes a 4-layer convolutional network with a, b, c, and d filters in each layer. $^\dagger$The algorithm was trained on both train and validation classes. We took the data from Dhillon et al. (2020) and Tian et al. (2020).

terms which converge to zero as the number of source training samples $m_c$, as well as the number of source classes $l$ increase.

## 5 EXPERIMENTS

In this section we experimentally analyze the neural collapse phenomenon and how it generalizes to new data points and new classes. We use reasonably good classifiers to demonstrate that, in addition to the neural collapse observed in training time by Papyan et al. (2020), it is also observable on test data from the same classes, as well as on data from new classes, as predicted by our theoretical results. We also show that, as expected intuitively, neural collapse is strongly correlated with accuracy in few-shot learning scenarios. The experiments are conducted over multiple datasets and multiple architectures, providing strong empirical evidence that neural collapse provides a compelling explanation for the good performance of foundation models in few-shot learning tasks. Experimental results are reported averaged over 20 random initialization together with 95% confidence intervals.

### 5.1 SETUP

**Method.** Following our theoretical setup, we first train a neural network classifier $\tilde{h} = \tilde{g} \circ f$ on a source task. Then we evaluate the few-shot performance of $f$ on target classification tasks by training a new classifier $h = g \circ f$, by minimizing the cross-entropy loss between its logits and the one-hot encodings of the labels. The training is conducted using SGD with learning rate $\eta$ and momentum 0.9 with batch size 64. Here, $\tilde{g}$ is the top linear layer of the neural network and $f$ is the mapping implemented by all other layers. At the second stage, given a target few-shot classification task with training data $S = \{(x_i, y_i)\}_{i=1}^n$, we train a new top layer $g$ as a solution of ridge regression acting on the dataset $\{(f(x_i), y_i)\}_{i=1}^n$ with regularization $\lambda_n = \alpha\sqrt{n}$. Thus, $g$ is a linear transformation with the weight matrix $w_{S,f} = (f(X)^\top f(X) + \lambda_n I)^{-1} f(X)^\top Y$, where $f(X)$ is the $n \times d$ data matrix for the ridge regression problem containing the feature vectors $\{f(x_i)\}$ (i.e., $f(X)^\top = [f(x_1), \ldots, f(x_n)]$) and $Y$ is the $n \times k$ label matrix (i.e., $Y^\top = [y_1, \ldots, y_n]$), where $X \in \mathbb{R}^{n \times d}$, $Y \in \mathbb{R}^{n \times k}$ and $f_\theta(X) \in \mathbb{R}^{n \times p}$. We did not apply any form of fine-tuning for $f$ at the second stage. In the experiments we sample 5-class classification tasks randomly from the target dataset, with $n_c$ training samples for each class (thus, altogether $n = 5n_c$ above), and measure the performance on 100 random test samples from each class. We report the resulting accuracy rates averaged over 100 randomly chosen tasks.

**Architectures and hyperparameters.** We experimented with two types of architectures for $\tilde{h}$: wide ResNets (Zagoruyko & Komodakis, 2016) and vanilla convolutional networks of the same structure without the residual connections, denoted by WRN-$N$-$M$ and Conv-$N$-$M$, where $N$ is the

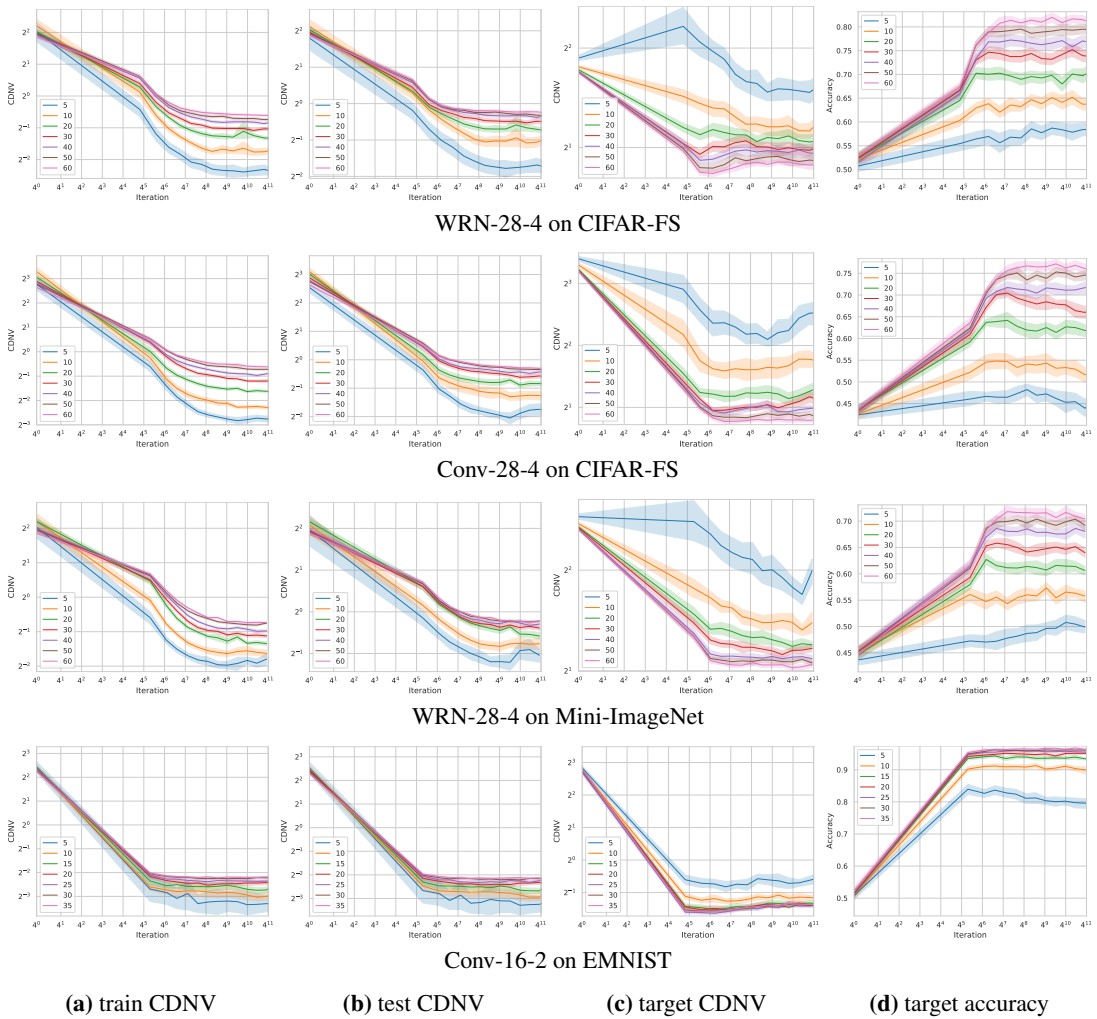

Figure 1: **Within-class variation collapse.** **(a)** CDNV on the source training data; **(b)** CDNV over the source test data; **(c)** CDNV over the target classes, all plotted in log-log scale. **(d)** Target accuracy rate (lin-log scale). In each experiment we trained the model on a different number of source classes $l \in \{5, 10, 20, 30, 40, 50, 60\}$ (as indicated in the legend).

depth and $M$ is the width factor. We used the following default hyperparameters: $\eta = 2^{-4}$, batch size 64 and $\alpha = 1$. In Figs. 2-3 of the appendix we provide experiments showing that the results are consistent for different $\eta$. We also provide experiments with a standard learning-rate schedule. See Appendix A for a complete description of the architectures.

**Datasets.** We consider four different datasets: (i) Mini-ImageNet (Vinyals et al., 2016); (ii) CIFAR-FS (Bertinetto et al., 2019); (iii) FC-100 (Oreshkin et al., 2018); and (iv) EMNIST (balanced) (Cohen et al., 2017). For a complete description of the datasets, see Appendix A.

## 5.2 RESULTS

While the ridge regression method to train the few-shot classifier $g$ may seem simplistic, it provides reasonably good performance, and hence it is suitable for studying the success of recent transfer/transductive learning methods (Dhillon et al., 2020; Tian et al., 2020) for few-shot learning. To demonstrate this, we compared the 1 and 5-shot performance of our simplistic method to several few-shot learning algorithms on Mini-ImageNet, CIFAR-FS and FC-100, summarized in Table 1. On each dataset, we report the average performance of our method on epochs between 90 and 100. As can be seen, the method we study in this paper is competitive with the rest of the literature on the

three benchmarks, especially in the 1-shot case (even achieving the state of the art on FC-100).[1] To improve the performance of our method a bit, we employed a standard learning rate scheduling with initial learning rate $\eta = 0.05$, decayed twice by a factor $0.1$ after 30 epochs each (accuracy rates are reported averaging over epochs 90–100, as before). Since the performance of these networks plateaued slightly after the first learning rate decay on the source test data, we also applied model selection based on this information, and used the network from the first 60 epochs (to avoid over-fitting to the source data and classes happening with the smallest learning rate) with the best source test performance. The combination of these two variations typically resulted in a small improvement of a few percentage points in the problems considered, see the last line of Table 1.

As our main experiment, we validate the theoretical assessments we made in Section 4. We argued that in the presence of neural collapse on the training classes, the trained feature map can be used for training a classifier with a small number of samples on new unseen classes. The argument asserts that if we observe neural collapse on a large set of source classes, then we expect to have neural collapse on new unseen classes as well, when assuming that the classes are selected in an i.i.d. manner. In this section we demonstrate that neural collapse generalizes to new samples from the same classes, and also to new classes, and we show that it is correlated with good few-shot performance.

To validate the above, we trained classifiers $\tilde{h}$ with a varying number of $l$ randomly selected source classes. For each run, we plot in Figure 1 the CDNV as a function of the epoch for the training and test datasets of the source classes and over the test samples of the target classes. In addition, we plot the 5-shot accuracy of ridge regression using the learned feature map $f$. Similar experiments with different numbers of target samples are reported in Figures 4 and 5 in the appendix.

As it can be seen in Figure 1, the CDNV decreases over the course of training on both training and test datasets of the source classes, showing that neural collapse generalizes to new samples from the training classes. Since the classification tasks with fewer number of source classes are easier to learn, the CDNV tends to be larger when training with a wider set of source classes. In contrast, we observe that when increasing the number of source classes, the presence of neural collapse in the target classes strengthens. This is in alignment with our theoretical expectations (the more "training" classes, the better generalization to new classes), and the few-shot performance also consistently improves when the overall number of source classes is increased. Note that in practice the CDNV does not decrease to zero (due to the limited sample sizes), and plateaus at a value above 1 on the target classes since we use a relatively small number of source classes (at most 60 here and 64 in the appendix). To validate the generality of our results, this phenomenon is demonstrated in several settings, e.g., using different network architectures and datasets in Figure 1. As can be seen, the values of the CDNV on the target classes are relatively large compared to those on the source classes, except for the results on EMNIST. However, these values still indicate a reasonable few-shot learning performance, as demonstrated in the experiments. These results consistently validate our theoretical findings, that is, that neural collapse generalizes to new source samples, it emerges for new classes, and its presence immediately facilitates good performance in few-shot learning. In Appendix B.2, we also demonstrate that similar phenomena can be observed for feature maps obtained from lower layers of the network, as well, although to a lesser extent.

## 6 CONCLUSIONS

Employing foundation models for transfer learning is a successful approach for dealing with over-fitting in the low-data regime. However, the reasons for this success are not clear. In this paper we presented a new perspective on this problem by connecting it to the newly discovered phenomenon of neural collapse. We showed that the within-class variance collapse tends to emerge in the test data associated with the classes encountered at train time and, more importantly, in new unseen classes when the new classes are drawn from the same distribution as the training classes. In addition, we showed that when neural collapse emerges in the new classes, then it requires very few samples to train a linear classifier on top of the learned feature representation that accurately predicts the new classes. These results provide a justification to the recent successes of transfer learning in few-shot tasks, as observed by Tian et al. (2020) and Dhillon et al. (2020).

---

[1]The inferior performance of our method compared to Distill-simple (Tian et al., 2020), is most likely due to the choice of the final few-shot classifier: our ridge regression classifier is inferior to their logistic regression solution (as shown in Table 1), while it is superior compared to their nearest neighbor classifier (see Table 5 in their paper).

ACKNOWLEDGEMENTS

During this work, Tomer Galanti was a research scientist intern at DeepMind. The authors would like to thank Csaba Szepesvári and Razvan Pascanu for illuminating discussions during the preparation of this manuscript, and Miruna Pîslar for her priceless technical support.

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

## A    EXPERIMENTAL DETAILS

**Datasets.**    Throughout the experiments, we consider four different datasets: (i) Mini-ImageNet (Vinyals et al., 2016) and (ii) CIFAR-FS (Bertinetto et al., 2019), (iii) FC-100 (Oreshkin et al., 2018) and EMNIST (balanced) (Cohen et al., 2017). Each dataset is split into meta-train, meta-validation and meta-test classes; we select the data for the source classes from the meta-training, and use similarly the meta-test data for the target tasks (we do not use the meta-validation classes). Each one of the class splits is also partitioned into train and test samples; we use these for training and evaluating our models. The Mini-ImageNet dataset contains 100 classes randomly chosen from ImageNet ILSVRC-2012 (Russakovsky et al., 2015) with 600 images of size $84 \times 84$ pixels per class. It is split into 64 meta-training classes, 16 meta-validation classes and 20 classes for meta-testing. CIFAR-FS and FC-100 are two derivatives of the CIFAR-100 dataset (Krizhevsky, 2012). CIFAR-FS consists of a random split of the CIFAR-100 classes into 64 classes for meta-training, 14 for meta-validation and 20 for meta-testing. FC-100 contains 100 classes which are grouped into 20 superclasses. These classes are partitioned into 60 meta-train classes from 12 superclasses, 20 classes from 4 superclasses for meta-validation, and 20 meta-test classes from 4 superclasses. We also consider the EMNIST dataset, which is an extension of the original MNIST dataset. We randomly split its classes into 35 source classes and 12 target classes (which are: 2, 10, 11, 12, 13, 16, 18, 22, 25, 33, 34, 44). We use random cropping augmentations at training time across all of the experiments.

**Architectures.**    We experimented with two types of architectures for $\tilde{h}$: wide ResNets (Zagoruyko & Komodakis, 2016) and vanilla convolutional networks. Wide ResNets start with a convolutional layer (with kernels $3 \times 3$ and 16 output channels), followed by three groups of layers. Each group of layers includes a convolutional layer, followed by a sequence of $N$ residual layers. Each residual layer in the $i$'th group contains two convolutional layers using $3 \times 3$ kernels and output $2^{i+3}M$ channels. Each convolutional layer is followed by a ReLU activations and batch normalization post activations. The network's penultimate layer is a mean pooling activation with kernels of size $4 \times 4$, returning an output of dimension 256 and followed by a linear layer. The vanilla convolutional networks have the same architectures as the wide ResNets, except that we omit the residual connections. The networks are denoted by WRN-$N$-$M$ and Conv-$N$-$M$, respectively.

## B    ADDITIONAL EXPERIMENTS

In this section we report additional experiments to provide further insights.

### B.1    VARYING HYPERPARAMETERS

We start by presenting experimental results to validate the consistency of our findings when varying different hyperparameters.

**Varying learning rates.**    We repeated the experiments in rows 1 and 4 of Figure 1 with learning rates $\eta = 2^{-2i-2}$ for $i = 1, 2, 3, 4$. We also report the train and test accuracy rates on the source data. The results are reported in Figure 2 and Figure 3. The results are consistent with the observations we had in Figure 1. Interestingly, when using smaller learning rates, the CDNV over the source training and test data tends to be smaller, while the CDNV on the target classes tends to be larger. We attribute this observation to overfitting to the source classes and indeed we observe that the few-shot performance is generally worse in those cases.

**Learning rate scheduling and varying number of target samples.**    To further demonstrate the relationship between neural collapse and generalization to new classes, we experimented with the default learning rate and standard learning rate scheduling (also used by Tian et al. (2020)), with varying number of target samples. In this experiment, we trained WRN-28-4 using SGD with the default learning rate and learning rate scheduling, starting with learning rate $\eta = 0.05$ with decay factor $0.1$ which is applied every 30 epochs twice. For the default training setting, in Figure 4(a-c) we report the dynamics of the CDNV on the source training data, the source test data (i.e., unseen test samples from the source classes), and the target classes (resp.) and in Figure 5(d-g) we plot the 1, 5, 10 and 20-shot accuracy rates of the network during training time. The results of learning rate scheduling are provided in Figure 5.

Similarly to Figure 1, as expected, we can observe that when increasing the number of source classes, the few-shot performance improves, while the CDNV on the target classes tends to decrease. Also, in line with our theory, the CDNV on the target classes is negatively correlated with the few-shot performance, that is, better neural collapse yields better performance. For example, in Figure 5(d-g) it is evident that the peak performance on all few-shot experiments for the case of 64 source classes is achieved around the minimal value of the CDNV on the target classes in Figure 5(c). This is achieved around iteration $16,000$ for CIFAR-FS and around iteration $20,000$ for Mini-ImageNet, a little bit after the first learning rate decay (at $15,000$ and $18,000$ steps, respectively). As can be seen, the performance slightly decreases after the peak iteration, while the CDNV on the target classes slightly increases, when the training starts to overfit to the source data and the source classes. This effect can be mitigated by selecting the final network based on the performance on the source test data, as we show in Table 1.

## B.2   NEURAL COLLAPSE AND LOWER LAYERS

We conducted a comparison between the behaviour of the penultimate layer (the top embedding layer) and a lower layer in the network, as the feature layer. We trained a WRN-28-4 on CIFAR-FS using the default hyperparameters as described in Section 2, but in this experiment we used the second-to-last embedding layer of the network as the feature layer for few-shot learning. A comparison of the behavior of this layer and the top embedding layer (that we use everywhere else in the paper) is shown in Figure 6 in terms of the CDNV on the source training data, the source test data, and the new classes (the target data), as well as for the 5-shot 5-class classification accuracy. As can be seen, the few-shot performance of the top embedding layer is superior to the performance of the lower layer. This is in agreement with the evidently smaller values of the CDNV given by the top embedding layer in comparison to the second-to-last embedding layer in all three cases (i.e., source train and test data and the target data). Nevertheless, we can see that the neural collapse phenomenon and the associated good few-shot performance is preserved in this lower layer of the network, however, less pronounced.

## B.3   DYNAMICS OF THE CLASS-EMBEDDING DISTANCES

In Section 4.1 we argued that the generalization bound in Proposition 1 is meaningful when the minimal distances $\min_{i \neq j} \|\mu_f(\tilde{S}_i) - \mu_f(\tilde{S}_j)\|^2$ (between the empirical class means) and $\min_{i \neq j} \|\mu_f(\tilde{P}_i) - \mu_f(\tilde{P}_j)\|^2$ (between the true class means) are not too small. Therefore, we empirically investigated their dynamics during training in our standard setting (WRN-28-4 with the default hyperparameters, see Section 2) on CIFAR-FS, considering a varying number $l \in \{5, 10, 20, 30, 40, 50, 60\}$ of source classes and learning rates $\eta \in \{2^{-2i-2}\}_{i=1}^{4}$. As can be seen in Figure 7, the values of $\min_{i \neq j} \|\mu_f(\tilde{S}_i) - \mu_f(\tilde{S}_j)\|^2$ and $\min_{i \neq j} \|\mu_f(\tilde{P}_i) - \mu_f(\tilde{P}_j)\|^2$ tend to increase during training. For completeness, we also plotted the values of $\min_{i \neq j} \|\mu_f(P_i) - \mu_f(P_j)\|^2$ for the target classes $\{P_c\}_{c=1}^{k}$ ($k = 20$ for CIFAR-FS). Interestingly, $\min_{i \neq j} \|\mu_f(P_i) - \mu_f(P_j)\|^2$ tends to grow with the number of source classes, showing improved generalization to new classes.

## B.4   CLASS-COVARIANCE NORMALIZED VARIANCE

In this section we consider the formal definition of Papyan et al. (2020) for neural collapse (their first definition), which uses a normalized variance definition somewhat different from CDNV we used in our analysis. For completeness, we present some experiments which demonstrate that our findings based on CDNV also apply to their normalization.

To give the formal definition, consider a training set $\tilde{S} = \bigcup_{c=1}^{l} \tilde{S}_c = \bigcup_{c=1}^{l} \{(\tilde{x}_{cj}, c)\}_{j=1}^{m_c}$ for an $l$-class classification problem, where $\tilde{S}_c$ is a collection of $m_c$ samples from class $c$. Assume that a neural network is trained to classify the samples in $\tilde{S}$, where after $t$ steps of the training we obtain the classifier $\tilde{h}_t = \tilde{g}_t \circ f_t$, where, as before, $\tilde{g}_t$ denotes the linear map of the last layer of the network, and $f_t$ is the feature map.

**Class-Covariance Normalized Variance.**   Papyan et al. (2020) define the *class-covariance normalized variance* (CCNV) to define neural collapse: For a given feature map $f$, let $\tilde{\mu}_c^f = \mu_f(\tilde{S}_c)$ denote the mean feature value of class $c$, and let $\tilde{\mu}_G^f = \text{Avg}_{c=1}^{l}[\tilde{\mu}_c^f]$ denote the global fea-

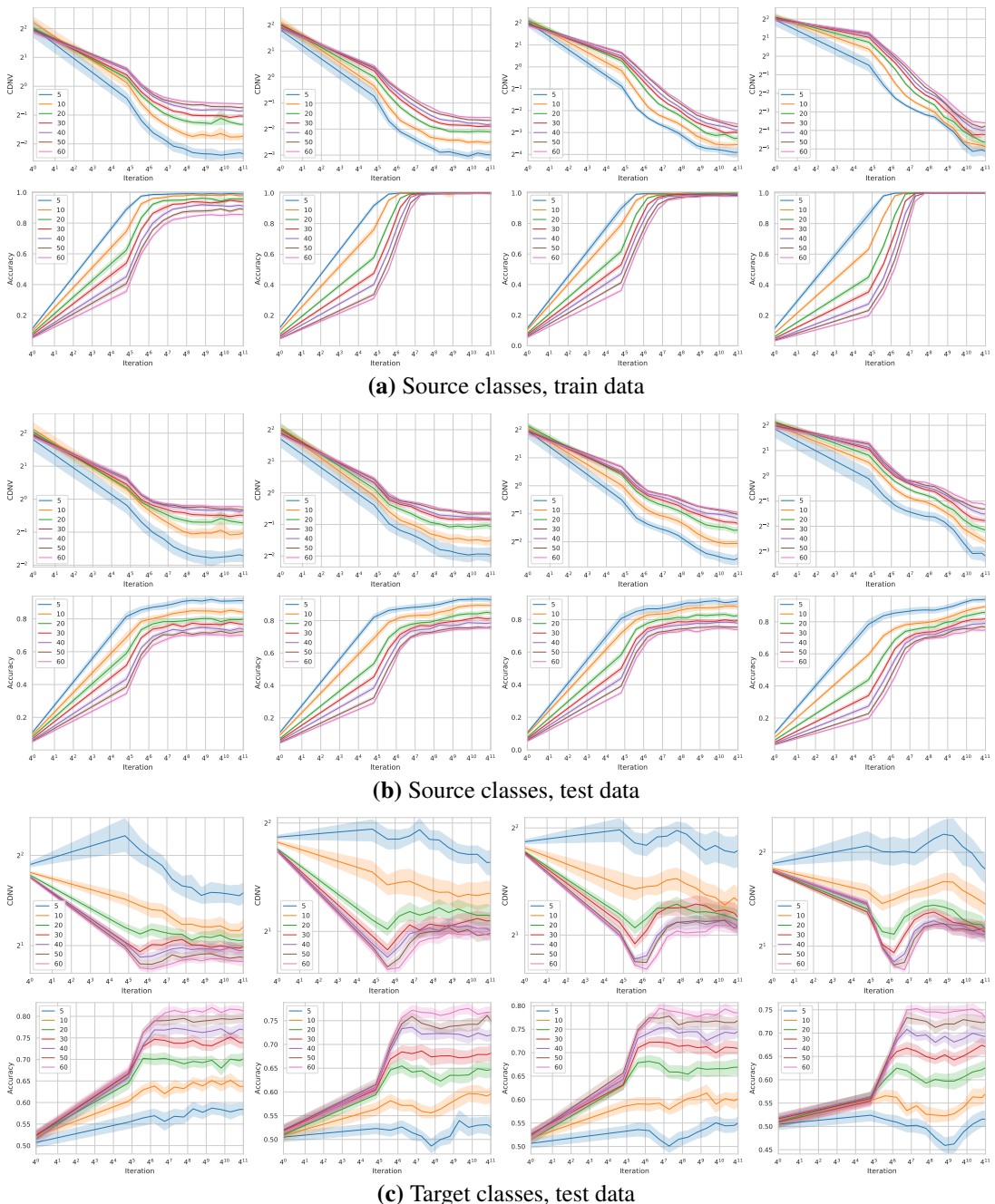

(a) Source classes, train data

(b) Source classes, test data

(c) Target classes, test data

Figure 2: **Averaged class variance and accuracy rates when varying the number of source classes on CIFAR-FS.** In **(a)** we plot the CDNV and accuracy rates on the source training data (resp.), in **(b)** on the source test data and in **(c)** on the target test data. In each experiment we trained a WRN-28-4 with SGD on a set of $l \in \{5, 10, 20, 30, 40, 50, 60\}$ source classes (as indicated in the legend). The $i$'th column stands for learning rate $\eta = 2^{-2i-2}$.

ture mean. The CCNV is defined to be $\mathrm{Tr}\left(\Sigma_W^f \Sigma_B^{f\,\dagger}\right)$, where $\Sigma_W^f$ and $\Sigma_B^f$ are the intra- and inter-class covariance matrices $\Sigma_W^f = \underset{\substack{c \in [l] \\ j \in [m_c]}}{\mathrm{Avg}}\left[(f(\tilde{x}_{cj}) - \tilde{\mu}_c^f) \cdot (f(\tilde{x}_{cj}) - \tilde{\mu}_c^f)^\top\right]$ and $\Sigma_B^f = \underset{c \in [l]}{\mathrm{Avg}}\left[(\tilde{\mu}_c^f - \tilde{\mu}_G^f) \cdot (\tilde{\mu}_c^f - \tilde{\mu}_G^f)^\top\right]$, where $A^+$ stands for the Moore-Penrose inverse of a (square)

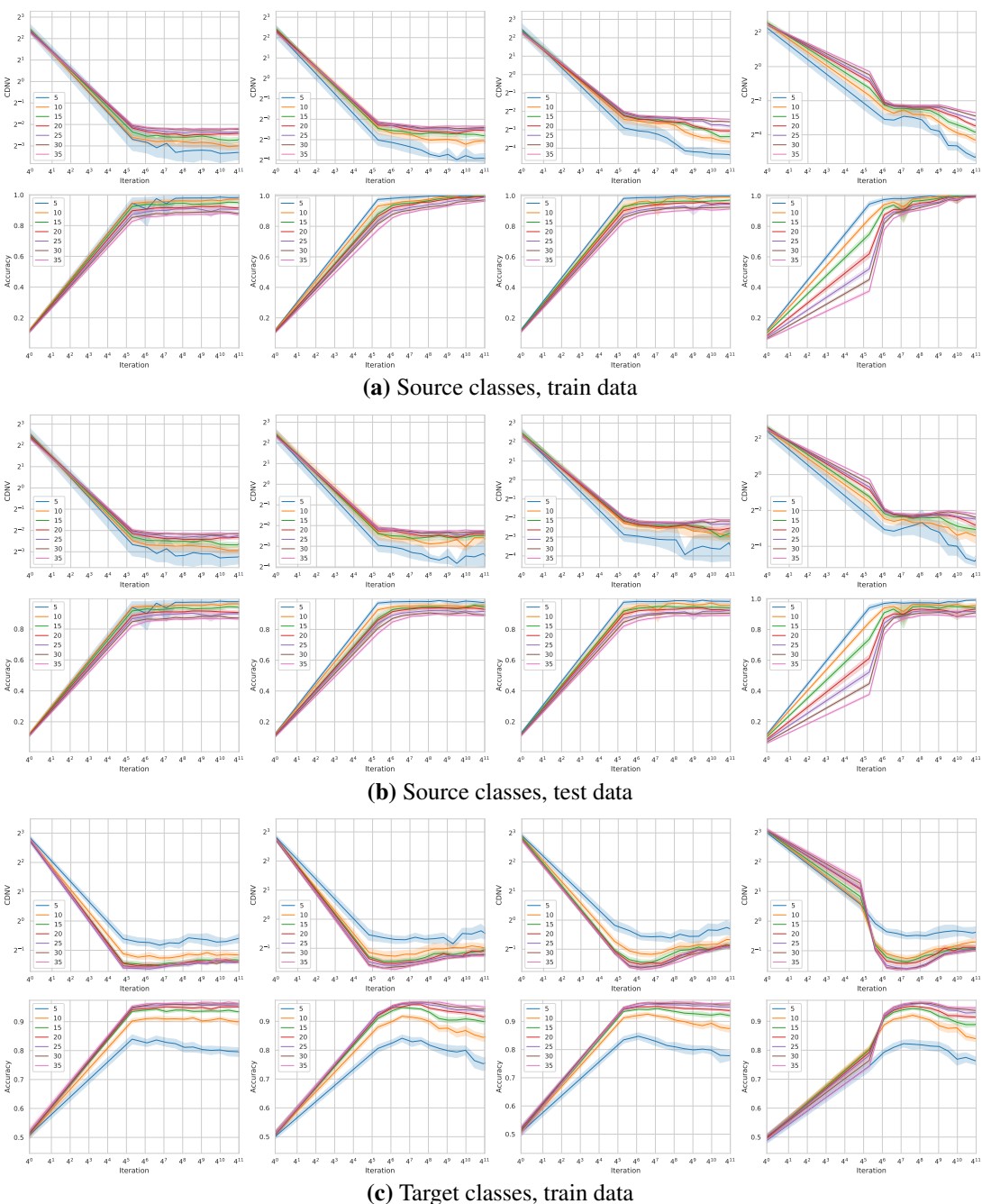

Figure 3: **Averaged class variance and accuracy rates when varying the number of source classes on EMNIST.** In **(a)** we plot the CDNV and accuracy rates on the source training data (resp.), in **(b)** on the source test data and in **(c)** on the target test data. In each experiment we trained a WRN-28-4 with SGD on a set of $l \in \{5, 10, 20, 30, 40, 50, 60\}$ source classes (as indicated in the legend). The $i$'th column stands for the results when training with learning rate $\eta = 2^{-2i-2}$.

matrix $A$. According to Papyan et al. (2020), (their first version of) neural collapse happens if $\lim_{t\to\infty} \text{Tr}\left(\Sigma_W^{f_t} \Sigma_B^{f_t \dagger}\right) = 0$.

In Figures 10 and 11 we plot the CCNV as a function of training iterations on the source training data, source test data and target classes. As can be seen, the value of the CCNV decreases on the

train and test data of the source classes. In addition, the CCNV on the target classes decreases when training with a larger set of source classes, which is due to better generalization. In contrast, the CCNV on the source classes increases when training with a larger set of source classes, since it increases the complexity of the optimization problem. These results are qualitatively very similar to the once presented for CDNV, showing that both definitions of neural collapse behaves essentially the same way. The corresponding train/test/few-shot accuracy rates are provided in Figures 2-3. In Figures 8-9 we provide the CDNV, CCNV and accuracy rates of training WRN-28-4 and Conv-28-4 on Mini-ImageNet and CIFAR-FS (resp.) with a varying number of source classes.

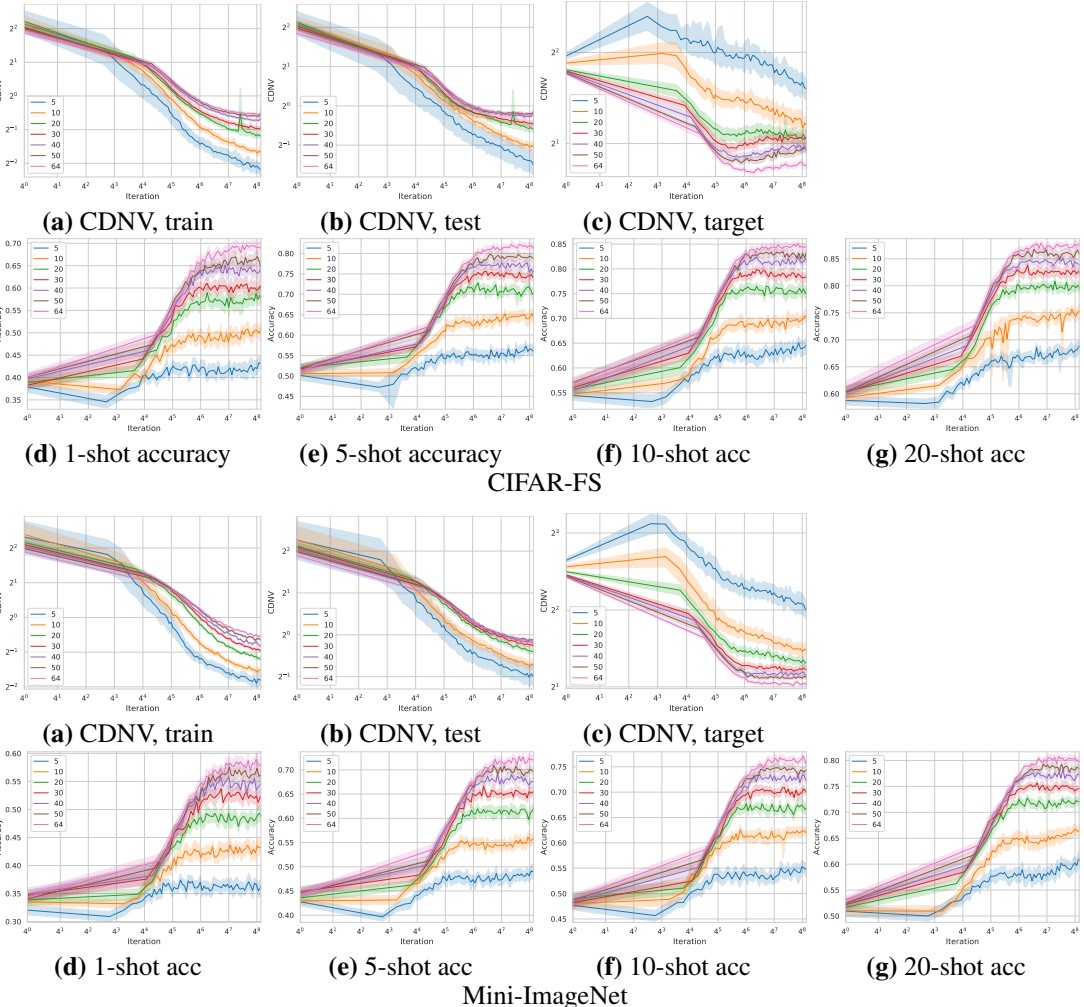

Figure 4: **Within-class variation and few-shot performance with learning rate** $\eta = 2^{-4}$**.** We trained WRN-28-4 using SGD with learning rate scheduling on $l \in \{5, 10, 20, 30, 40, 50, 64\}$ source classes (as indicated in the legend). For each dataset, in **(a-c)** we plot the CDNV on the train and test data and the target classes (resp.). In **(d-g)** we plot the 1,5,10 and 20-shot accuracy rates (resp.).

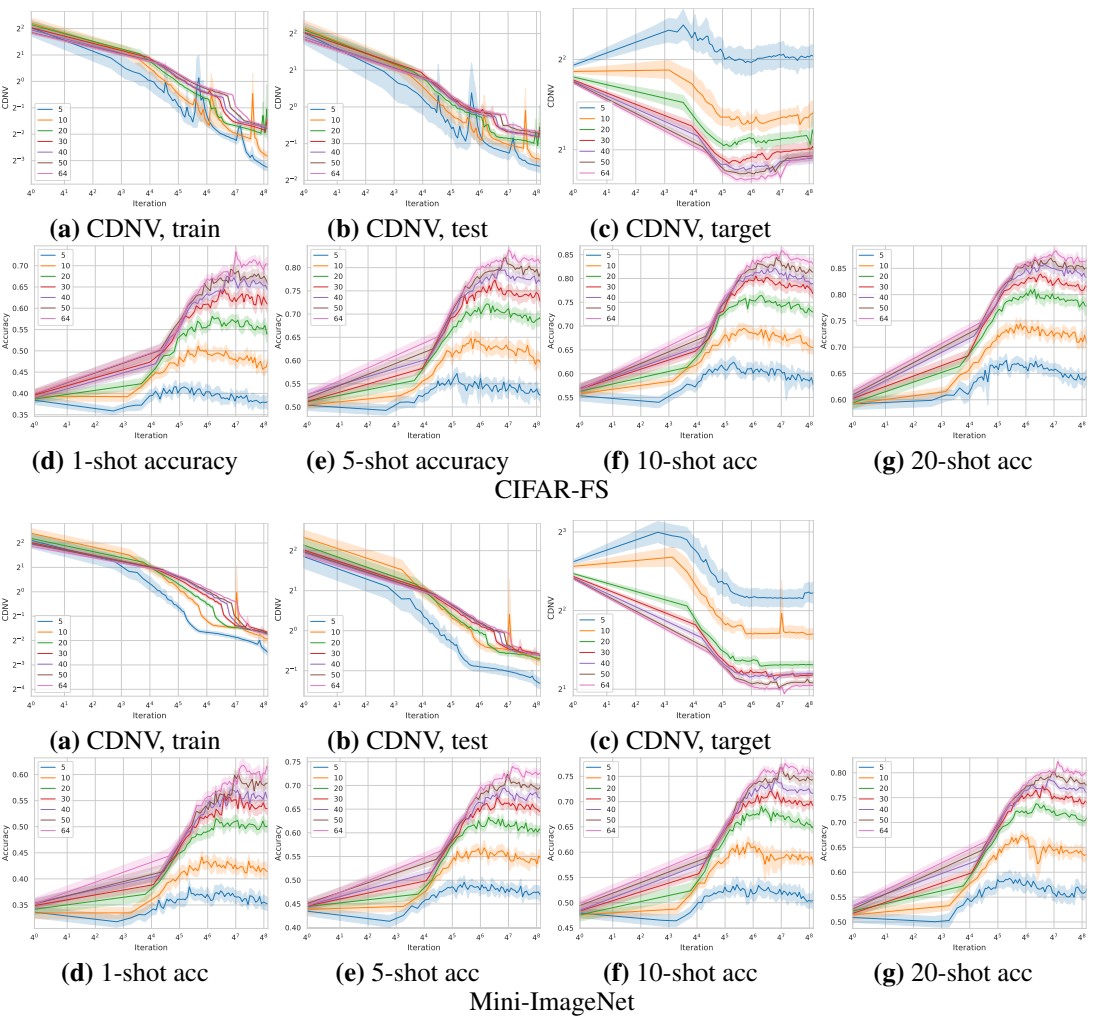

**(a)** CDNV, train     **(b)** CDNV, test     **(c)** CDNV, target

**(d)** 1-shot accuracy    **(e)** 5-shot accuracy    **(f)** 10-shot acc    **(g)** 20-shot acc

CIFAR-FS

**(a)** CDNV, train     **(b)** CDNV, test     **(c)** CDNV, target

**(d)** 1-shot acc    **(e)** 5-shot acc    **(f)** 10-shot acc    **(g)** 20-shot acc

Mini-ImageNet

Figure 5: **Within-class variation and few-shot performance with learning rate scheduling.** We trained WRN-28-4 using SGD with learning rate scheduling on $l \in \{5, 10, 20, 30, 40, 50, 64\}$ source classes (as indicated in the legend). For each dataset, in **(a-c)** we plot the CDNV on the train and test data and the target classes (resp.). In **(d-g)** we plot the 1,5,10 and 20-shot accuracy rates (resp.).

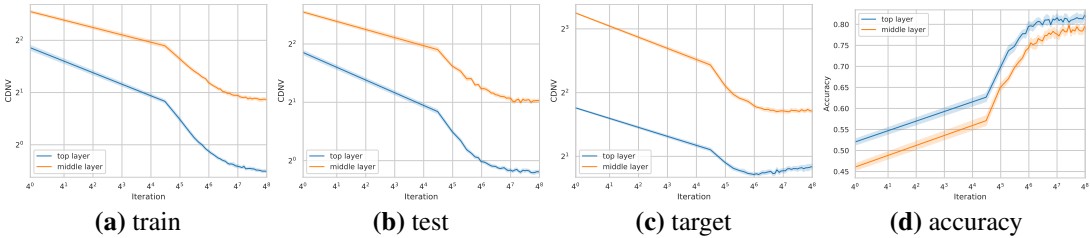

**(a)** train      **(b)** test      **(c)** target      **(d)** accuracy

Figure 6: **Within-class variation collapse of the second-to-last embedding layer.** In each experiment we trained a WRN-28-4 using SGD with $\eta = 2^{-4}$ on a set of $l = 64$ source classes on CIFAR-FS. We compare the CDNVs and 5-shot accuracy rates of the second-to-last embedding layer and the top embedding layer of the network. In **(a)** we compare the CDNV on the source train data, in **(b)** the CDNV on the source test data, **(c)** the CDNV on the target classes and in **(d)** the 5-shot accuracy rates.

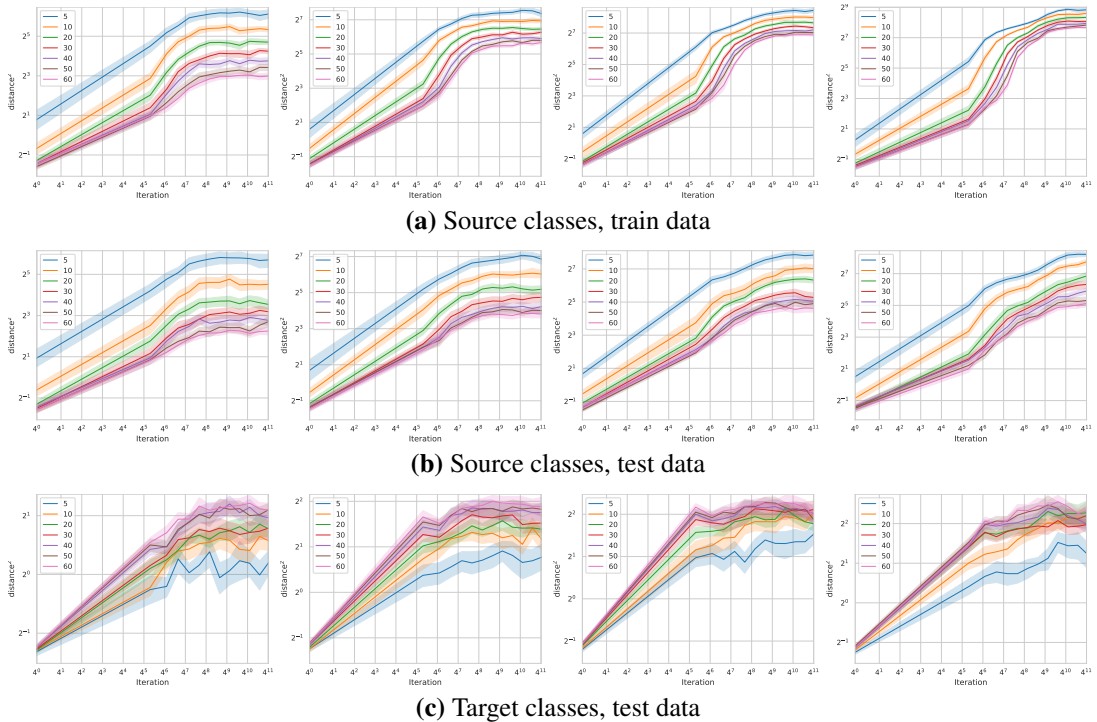

Figure 7: **Dynamics of minimal class-means distance.** In (a) we plot $\min_{i \neq j} \|\mu_f(\tilde{S}_i) - \mu_f(\tilde{S}_j)\|$, in (b) we plot $\min_{i \neq j} \|\mu_f(\tilde{P}_i) - \mu_f(\tilde{P}_j)\|$ and in (c) we plot $\min_{i \neq j} \|\mu_f(P_i) - \mu_f(P_j)\|$ as a function of the number of training iterations. In each experiment we trained a WRN-28-4 using SGD on a set of $l \in \{5, 10, 20, 30, 40, 50, 60\}$ source classes on CIFAR-FS (as indicated in the legend). The $i$'th column corresponds to the results of training with SGD with $\eta = 2^{-2i-2}$.

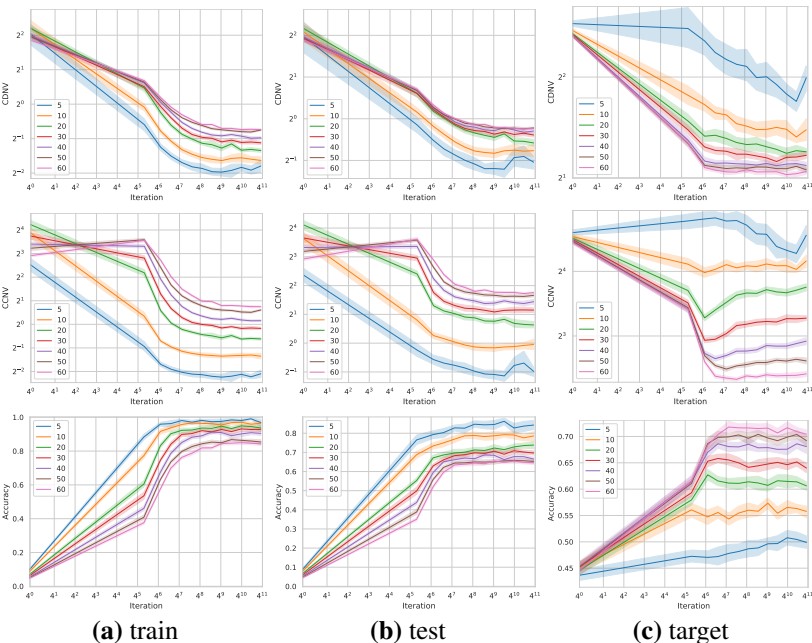

Figure 8: **Results of WRN-28-4 on Mini-ImageNet. (row 1)** CDNV, **(row 2)** CCNV, and **(row 3)** accuracy on the source train, test, and the target data (columns a,b,c, resp.). Each model was trained using SGD with $\eta = 2^{-4}$ on a set of $l \in \{5, 10, 20, 30, 40, 50, 60\}$ source classes (as indicated in the legend).

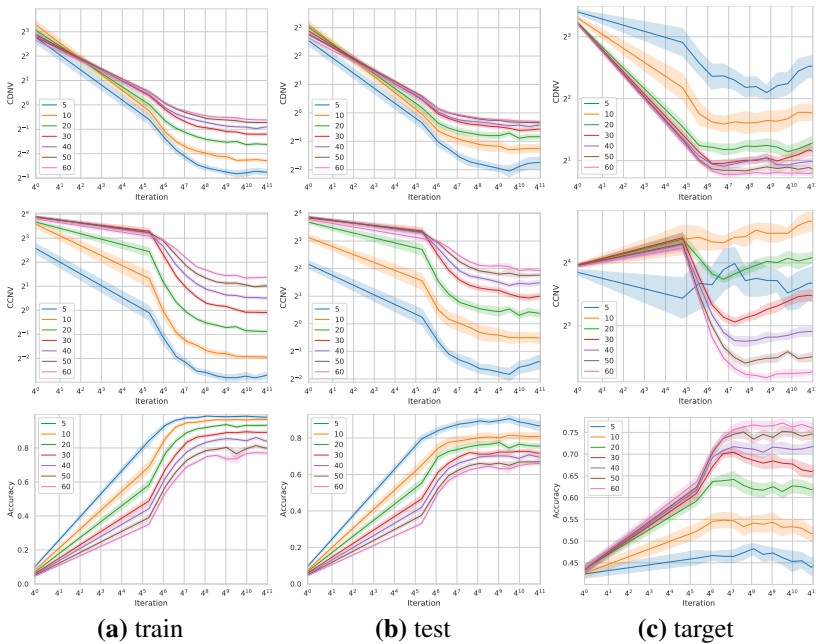

Figure 9: **Results of Conv-28-4 on CIFAR-FS. (row 1)** CDNV, **(row 2)** CCNV, and **(row 3)** accuracy on the source train, test, and the target data (columns a,b,c, resp.). Each model was trained using SGD with $\eta = 2^{-4}$ on a set of $l \in \{5, 10, 20, 30, 40, 50, 60\}$ source classes (as indicated in the legend).

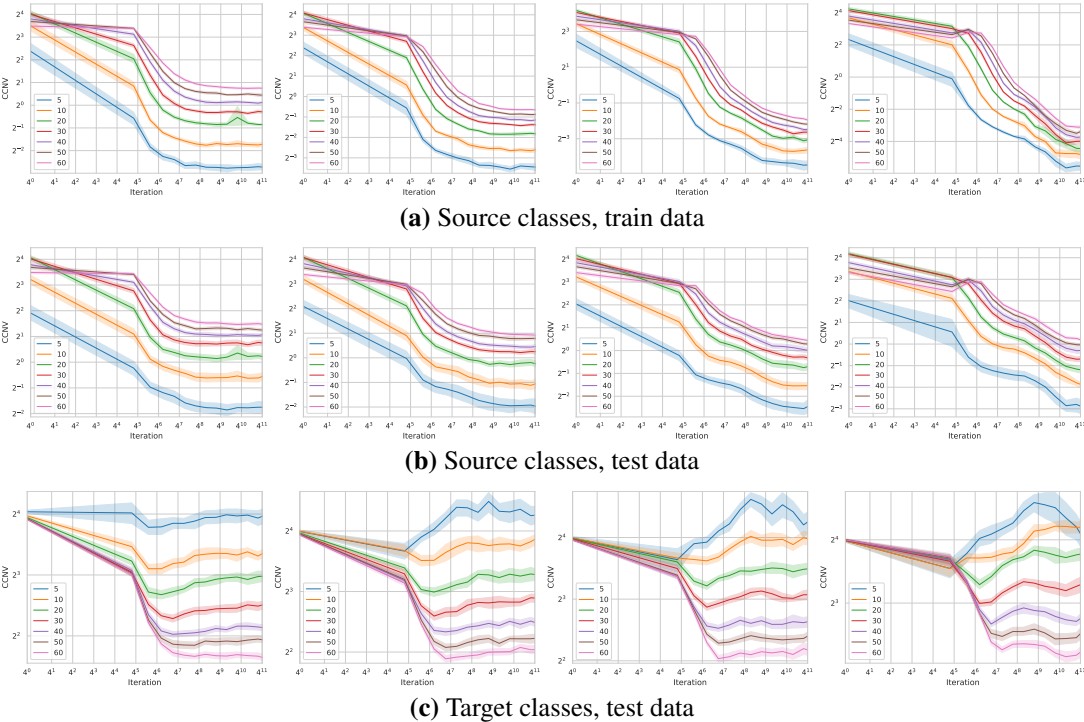

Figure 10: **Within-class variation collapse on CIFAR-FS.** In **(a)** We plot the CCNV on the source training data, in **(b)** on the source test data and in **(c)** on the target test data. In each experiment we trained a WRN-28-4 using SGD on a set of $l \in \{5, 10, 20, 30, 40, 50, 60\}$ source classes (as indicated in the legend). The $i$'th column stands for the results when training with learning rate $\eta = 2^{-2i-2}$.

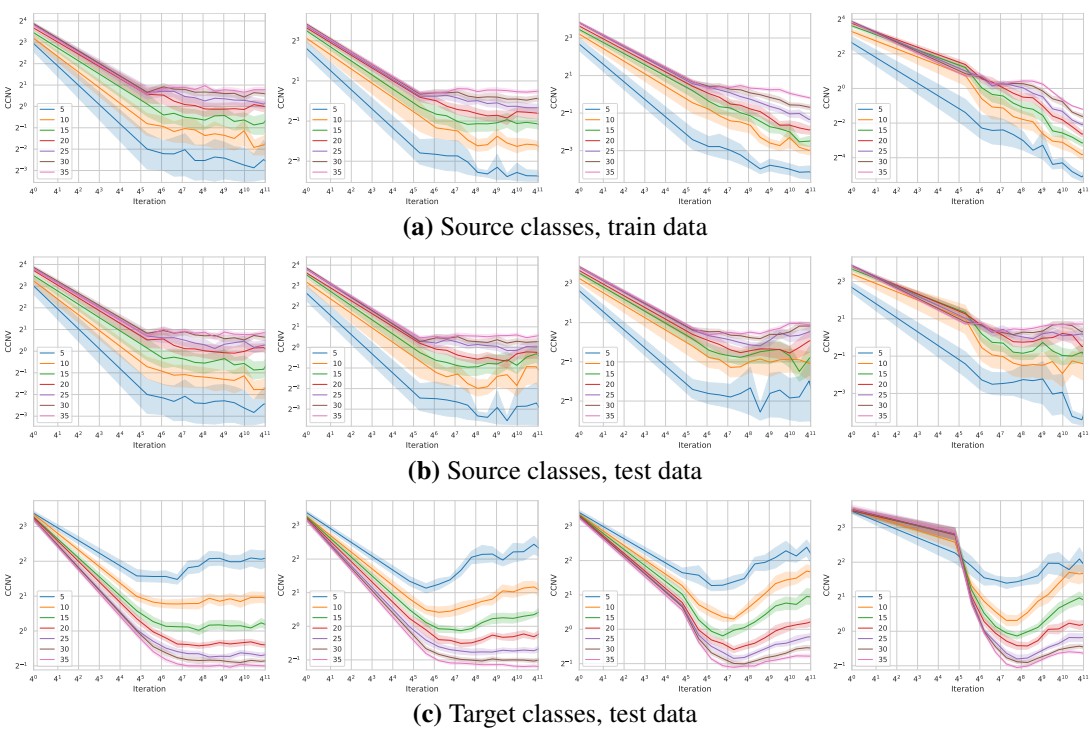

Figure 11: **Within-class variation collapse on EMNIST.** In **(a)** We plot the CCNV on the source training data, in **(b)** on the source test data and in **(c)** on the target test data. We trained a WRN-28-4 with SGD on a set of $l \in \{5, 10, 15, 20, 25, 30, 35\}$ source classes (as indicated in the legend). The $i$'th column shows results for learning rate $\eta = 2^{-2i-2}$.

## C  PROOF OF PROPOSITION 1

**Lemma 1.** *Let $\tilde{P}_c$ be a class distribution, $\delta \in (0,1)$ and $\tilde{S}_c = \{\tilde{x}_{cj}\}_{j=1}^{m_c} \sim \tilde{P}_c^{m_c}$ be a dataset of $m_c$ samples. Let $f$ be the output of the learning algorithm. If*

$$
\begin{aligned}
\left\| \mathbb{E}_{x \sim \tilde{P}_c}[f(x)] - \mathrm{Avg}_{x \in \tilde{S}_c}[f(x)] \right\| &\leq \epsilon_1^c(\delta); \\
\left| \mathbb{E}_{x \sim \tilde{P}_c}[\|f(x)\|^2] - \mathrm{Avg}_{x \in \tilde{S}_c}[\|f(x)\|^2] \right| &\leq \epsilon_2^c(\delta),
\end{aligned}
\tag{2}
$$

*then we have*

$$
\mathrm{Var}_f(\tilde{P}_c) \leq \mathrm{Var}_f(\tilde{S}_c) + \epsilon_2^c(\delta) + 2\|\mu_f(\tilde{P}_c)\| \cdot \epsilon_1^c(\delta) + \epsilon_1^c(\delta)^2 .
\tag{3}
$$

*Proof.* Recall that

$$
\mathrm{Var}_f(\tilde{P}_c) = \mathbb{E}_{x \sim \tilde{P}_c}[\|f(x)\|^2] - \|\mathbb{E}_{x \sim \tilde{P}_c}[f(x)]\|^2 .
\tag{4}
$$

We would like to upper bound $\mathbb{E}_{x \sim \tilde{P}_c}[\|f(x)\|^2]$ in terms of $\mathrm{Avg}_{x \in \tilde{S}_c}[\|f(x)\|^2]$ and to lower bound $\|\mathbb{E}_{x \sim \tilde{P}_c}[f(x)]\|^2$ using $\|\mathrm{Avg}_{x \in \tilde{S}_c}[f(x)]\|^2$. By (2),

$$
\mathbb{E}_{x \sim \tilde{P}_c}[\|f(x)\|^2] \leq \mathrm{Avg}_{j=1}^{m_c}[\|f(\tilde{x}_{cj})\|^2] + \epsilon_2^c(\delta) .
\tag{5}
$$

By the triangle inequality,

$$
\|\mathbb{E}_{x \sim \tilde{P}_c}[f(x)]\| + \left\| \mathbb{E}_{x \sim \tilde{P}_c}[f(x)] - \mathrm{Avg}_{j=1}^{m_c}[f(\tilde{x}_{cj})] \right\| \geq \|\mathrm{Avg}_{j=1}^{m_c}[f(\tilde{x}_{cj})]\| .
$$

By (2),

$$
\left\| \mathbb{E}_{x \sim \tilde{P}_c}[f(x)] - \mathrm{Avg}_{j=1}^{m_c}[f(\tilde{x}_{cj})] \right\| \leq \epsilon_1^c(\delta) .
$$

Hence,

$$
\|\mathbb{E}_{x \sim \tilde{P}_c}[f(x)]\| + \epsilon_1^c(\delta) \geq \|\mathrm{Avg}_{j=1}^{m_c}[f(\tilde{x}_{cj})]\| ,
$$

which implies

$$
\left\| \mathrm{Avg}_{j=1}^{m_c}[f(\tilde{x}_{cj})] \right\|^2 \leq \left\| \mathbb{E}_{x \sim \tilde{P}_c}[f(x)] \right\|^2 + 2\epsilon_1^c(\delta) \cdot \|\mathbb{E}_{x \sim \tilde{P}_c}[f(x)]\| + \epsilon_1^c(\delta)^2 .
\tag{6}
$$

Combining (5) and (6), we obtain

$$
\begin{aligned}
\mathrm{Var}_f(\tilde{P}_c) &\leq \mathrm{Var}_f(\tilde{S}_c) + \epsilon_2^c(\delta) + 2 \left\| \mathbb{E}_{x \sim \tilde{P}_c}[f(x)] \right\| \cdot \epsilon_1^c(\delta) + \epsilon_1^c(\delta)^2 \\
&= \mathrm{Var}_f(\tilde{S}_c) + \epsilon_2^c(\delta) + 2\|\mu_f(\tilde{P}_c)\| \cdot \epsilon_1^c(\delta) + \epsilon_1^c(\delta)^2 ,
\end{aligned}
$$

completing the proof. □

**Proposition 1.** *Fix two source classes, $i$ and $j$ with distributions $\tilde{P}_i$ and $\tilde{P}_j$, and let $\delta \in (0,1)$. Let $\tilde{S}_c \sim \tilde{P}_c^{m_c}$ for $c \in \{i,j\}$. Let*

$$
A = \frac{\epsilon_1^i(\delta/4) + \epsilon_1^j(\delta/4)}{\|\mu_f(\tilde{P}_i) - \mu_f(\tilde{P}_j)\|} \quad \text{and} \quad B = \frac{\mathrm{Avg}_{c \in \{i,j\}}\left[ \epsilon_2^c(\delta/4) + 2\|\mu_f(\tilde{P}_c)\| \cdot \epsilon_1^c(\delta/4) + \epsilon_1^c(\delta/4)^2 \right]}{\|\mu_f(\tilde{S}_i) - \mu_f(\tilde{S}_j)\|^2} .
$$

*Then, with probability at least $1 - \delta$ over $\tilde{S}$, we have $V_f(\tilde{P}_i, \tilde{P}_j) \leq \left( V_f(\tilde{S}_i, \tilde{S}_j) + B \right) (1 + A)^2$.*

*Proof.* By definition and the union bound, with probability at least $1 - \delta$, the following inequalities hold simultaneously for all $c \in \{i,j\}$:

$$
\begin{aligned}
\left\| \mathbb{E}_{x \sim \tilde{P}_c}[f(x)] - \mathrm{Avg}_{x \in \tilde{S}_c}[f(x)] \right\| &\leq \epsilon_1^c(\delta/4); \\
\left| \mathbb{E}_{x \sim \tilde{P}_c}[\|f(x)\|^2] - \mathrm{Avg}_{x \in \tilde{S}_c}[\|f(x)\|^2] \right| &\leq \epsilon_2^c(\delta/4) .
\end{aligned}
\tag{7}
$$

In the rest of the proof, we assume that the above inequalities hold. Let $\Delta = \left| \|\mu_f(\tilde{P}_i) - \mu_f(\tilde{P}_j)\|^2 - \|\mu_f(\tilde{S}_i) - \mu_f(\tilde{S}_j)\|^2 \right|$. By simple algebraic manipulations,

$$
\begin{aligned}
\frac{1}{\|\mu_f(\tilde{P}_i) - \mu_f(\tilde{P}_j)\|^2} &= \frac{1}{\|\mu_f(\tilde{S}_i) - \mu_f(\tilde{S}_j)\|^2} - \frac{\|\mu_f(\tilde{P}_i) - \mu_f(\tilde{P}_j)\|^2 - \|\mu_f(\tilde{S}_i) - \mu_f(\tilde{S}_j)\|^2}{\|\mu_f(\tilde{P}_i) - \mu_f(\tilde{P}_j)\|^2 \cdot \|\mu_f(\tilde{S}_i) - \mu_f(\tilde{S}_j)\|^2} \\
&\leq \frac{1}{\|\mu_f(\tilde{S}_i) - \mu_f(\tilde{S}_j)\|^2} + \frac{\Delta}{\|\mu_f(\tilde{P}_i) - \mu_f(\tilde{P}_j)\|^2 \cdot \|\mu_f(\tilde{S}_i) - \mu_f(\tilde{S}_j)\|^2} .
\end{aligned}
\tag{8}
$$

By (7) and Lemma 1 we have

$$\forall c \in \{i,j\}: \ \mathrm{Var}_f(\tilde{P}_c) \ \leq \ \mathrm{Var}_f(\tilde{S}_c) + \epsilon_2^c(\delta/4) + 2\|\mu_f(\tilde{P}_c)\| \cdot \epsilon_1^c(\delta/4) + \epsilon_1^c(\delta/4)^2 \ . \tag{9}$$

Multiplying both sides of (8) by $\mathrm{Var}_f(\tilde{P}_c)$, and using (9), combining the above inequality with (8), we obtain

$$
\begin{aligned}
\frac{\mathrm{Var}_f(\tilde{P}_c)}{\|\mu_f(\tilde{P}_i) - \mu_f(\tilde{P}_j)\|^2} \ \leq \ & \frac{\mathrm{Var}_f(\tilde{S}_c)}{\|\mu_f(\tilde{S}_i) - \mu_f(\tilde{S}_j)\|^2} + \frac{\mathrm{Var}_f(\tilde{S}_c) \cdot \Delta}{\|\mu_f(\tilde{P}_i) - \mu_f(\tilde{P}_j)\|^2 \cdot \|\mu_f(\tilde{S}_i) - \mu_f(\tilde{S}_j)\|^2} \\
& + \frac{\left(\epsilon_2^c(\delta/4) + 2\|\mu_f(\tilde{P}_c)\| \cdot \epsilon_1^c(\delta/4) + \epsilon_1^c(\delta/4)^2\right) \cdot \Delta}{\|\mu_f(\tilde{P}_i) - \mu_f(\tilde{P}_j)\|^2 \cdot \|\mu_f(\tilde{S}_i) - \mu_f(\tilde{S}_j)\|^2} \\
& + \frac{\epsilon_2^c(\delta/4) + 2\|\mu_f(\tilde{P}_c)\| \cdot \epsilon_1^c(\delta/4) + \epsilon_1^c(\delta/4)^2}{\|\mu_f(\tilde{S}_i) - \mu_f(\tilde{S}_j)\|^2} \ .
\end{aligned}
\tag{10}
$$

Averaging (10) over $c \in \{i,j\}$ gives

$$
\begin{aligned}
V_f(\tilde{P}_i, \tilde{P}_j) \ \leq \ & V_f(\tilde{S}_i, \tilde{S}_j) \left(1 + \frac{\Delta}{\|\mu_f(\tilde{P}_i) - \mu_f(\tilde{P}_j)\|^2}\right) \\
& + \frac{\mathrm{Avg}_{c \in \{i,j\}} \left[\epsilon_2^c(\delta/4) + 2\|\mu_f(\tilde{P}_c)\| \cdot \epsilon_1^c(\delta/4) + \epsilon_1^c(\delta/4)^2\right] \cdot \Delta}{\|\mu_f(\tilde{P}_i) - \mu_f(\tilde{P}_j)\|^2 \cdot \|\mu_f(\tilde{S}_i) - \mu_f(\tilde{S}_j)\|^2} \\
& + \frac{\mathrm{Avg}_{c \in \{i,j\}} \left[\epsilon_2^c(\delta/4) + 2\|\mu_f(\tilde{P}_c)\| \cdot \epsilon_1^c(\delta/4) + \epsilon_1^c(\delta/4)^2\right]}{\|\mu_f(\tilde{S}_i) - \mu_f(\tilde{S}_j)\|^2} \ .
\end{aligned}
\tag{11}
$$

By the triangle inequality and (7),

$$
\begin{aligned}
\|\mu_f(\tilde{S}_i) - \mu_f(\tilde{S}_j)\| \ & \leq \ \|\mu_f(\tilde{P}_i) - \mu_f(\tilde{P}_j)\| + \|\mu_f(\tilde{S}_i) - \mu_f(\tilde{P}_i)\| + \|\mu_f(\tilde{S}_j) - \mu_f(\tilde{P}_j)\| \\
& \leq \ \|\mu_f(\tilde{P}_i) - \mu_f(\tilde{P}_j)\| + \epsilon_1^i(\delta/4) + \epsilon_1^j(\delta/4),
\end{aligned}
$$

and by a symmetric argument,

$$\left|\|\mu_f(\tilde{P}_i) - \mu_f(\tilde{P}_j)\| - \|\mu_f(\tilde{S}_i) - \mu_f(\tilde{S}_j)\|\right| \ \leq \ \epsilon_1^i(\delta/4) + \epsilon_1^j(\delta/4) \ .$$

This and (7) imply

$$
\begin{aligned}
\Delta \ & = \ \left|\|\mu_f(\tilde{P}_i) - \mu_f(\tilde{P}_j)\|^2 - \|\mu_f(\tilde{S}_i) - \mu_f(\tilde{S}_j)\|^2\right| \\
& = \ \left|\|\mu_f(\tilde{P}_i) - \mu_f(\tilde{P}_j)\| - \|\mu_f(\tilde{S}_i) - \mu_f(\tilde{S}_j)\|\right| \left(\|\mu_f(\tilde{P}_i) - \mu_f(\tilde{P}_j)\| + \|\mu_f(\tilde{S}_i) - \mu_f(\tilde{S}_j)\|\right) \\
& \leq \ \left(\epsilon_1^i(\delta/4) + \epsilon_1^j(\delta/4)\right) \cdot \left(2\|\mu_f(\tilde{P}_i) - \mu_f(\tilde{P}_j)\| + \epsilon_1^i(\delta/4) + \epsilon_1^j(\delta/4)\right) \ .
\end{aligned}
$$

Plugging the above bound into (11) shows that, with probability at least $1 - \delta$,

$$
\begin{aligned}
V_f(\tilde{P}_i, \tilde{P}_j) \ \leq \ & V_f(\tilde{S}_i, \tilde{S}_j) \left(1 + \frac{\epsilon_1^i(\delta/4) + \epsilon_1^j(\delta/4)}{\|\mu_f(\tilde{P}_i) - \mu_f(\tilde{P}_j)\|} + \left[\frac{\epsilon_1^i(\delta/4) + \epsilon_1^j(\delta/4)}{\|\mu_f(\tilde{P}_i) - \mu_f(\tilde{P}_j)\|}\right]^2\right) \\
& + \frac{2\,\mathrm{Avg}_{c \in \{i,j\}} \left[\epsilon_2^c(\delta/4) + 2\|\mu_f(\tilde{P}_c)\| \cdot \epsilon_1^c(\delta/4) + \epsilon_1^c(\delta/4)^2\right] \cdot \left(\epsilon_1^i(\delta/4) + \epsilon_1^j(\delta/4)\right)}{\|\mu_f(\tilde{P}_i) - \mu_f(\tilde{P}_j)\| \cdot \|\mu_f(\tilde{S}_i) - \mu_f(\tilde{S}_j)\|^2} \\
& + \frac{\mathrm{Avg}_{c \in \{i,j\}} \left[\epsilon_2^c(\delta/4) + 2\|\mu_f(\tilde{P}_c)\| \cdot \epsilon_1^c(\delta/4) + \epsilon_1^c(\delta/4)^2\right] \cdot \left(\epsilon_1^i(\delta/4) + \epsilon_1^j(\delta/4)\right)^2}{\|\mu_f(\tilde{P}_i) - \mu_f(\tilde{P}_j)\|^2 \cdot \|\mu_f(\tilde{S}_i) - \mu_f(\tilde{S}_j)\|^2} \\
& + \frac{\mathrm{Avg}_{c \in \{i,j\}} \left[\epsilon_2^c(\delta/4) + 2\|\mu_f(\tilde{P}_c)\| \cdot \epsilon_1^c(\delta/4) + \epsilon_1^c(\delta/4)^2\right]}{\|\mu_f(\tilde{S}_i) - \mu_f(\tilde{S}_j)\|^2} \\
\ \leq \ & \left(V_f(\tilde{S}_i, \tilde{S}_j) + B\right)(1 + A)^2 \ ,
\end{aligned}
$$

completing the proof. $\qquad\square$

# D  PROOF OF PROPOSITIONS 2

**Proposition 2.** *Let $\mathcal{F}^* \subset \mathcal{F}$ be any finite set of functions with $\Delta(\mathcal{F}^*) = \inf_{f \in \mathcal{F}^*} \inf_{P_c \neq P_{c'}} \|\mu_f(P_c) - \mu_f(P_{c'})\| > 0$. Then, with probability at least $1 - \delta$ over the selection of source class distributions $\tilde{\mathcal{P}}$,*

$$\mathbb{E}_{P_c \neq P_{c'}}[V_f(P_c, P_{c'})] \leq \mathrm{Avg}_{i \neq j}\Big[V_f(\tilde{P}_i, \tilde{P}_j)\Big] + \left(8 + \frac{16 \sup\limits_{f \in \mathcal{F}^*, P' \in \mathcal{C}} \mathrm{Var}_f(P')}{\Delta(\mathcal{F}^*)}\right) \cdot \frac{\sqrt{2\pi \log(l)} \mathbb{E}[R(H_{\mathcal{F}^*}(\tilde{\mathcal{P}}))]}{(l-1) \cdot \Delta(\mathcal{F}^*)^2}$$

$$+ \left(1 + \frac{4 \sup_{x \in \mathcal{X}, f \in \mathcal{F}^*} \|f(x)\|}{\Delta(\mathcal{F}^*)}\right) \cdot \frac{2\sqrt{\log(1/\delta)} \cdot \sup\limits_{f \in \mathcal{F}^*, P' \in \mathcal{C}} \mathrm{Var}_f(P')}{\sqrt{l} \cdot \Delta(\mathcal{F}^*)^2} \ .$$

To prove this proposition, we apply Theorem 2 of Maurer & Pontil (2019). In order to do so, we need the following definitions (their Definition 1): Let $\mathcal{U}$ be any set and $A : \mathcal{U}^l \to \mathbb{R}$. For $\mathbf{z} = (z_1, \dots, z_l), j \in \{1, \dots, l\}$ and $z_j' \in \mathcal{U}$, we define the $j$-th partial difference operator on $A$ as

$$D_{z_j'}^j A(\mathbf{z}) \ = \ A(\dots, z_{j-1}, z_j, z_{j+1} \dots) - A(\dots, z_{j-1}, z_j', z_{j+1}, \dots) \ .$$

In addition, we denote

$$M(A) \ = \ \max_{j \in [l]} \sup_{\substack{\mathbf{z} \in \mathcal{U}^l \\ z_j' \in \mathcal{U}: \ z_j' \neq z_j}} \frac{D_{z_j'}^j A(\mathbf{z})}{\|z_j - z_j'\|} \ ,$$

and

$$J(A) \ = \ l \cdot \max_{r \neq j \in [l]} \sup_{\substack{\mathbf{z} \in \mathcal{U}^l \\ z_j' \in \mathcal{U}: \ z_j' \neq z_j \\ z_r' \in \mathcal{U}: \ z_r' \neq z_r}} \frac{D_{z_r'}^r D_{z_j'}^j A(\mathbf{z})}{\|z_j - z_j'\|} \ ,$$

and the maximal difference

$$K(A) \ = \ \max_{j \in [l]} \sup_{\substack{\mathbf{z} \in \mathcal{U}^l \\ z_j' \in \mathcal{U}: \ z_j' \neq z_j}} D_{z_j'}^j A(\mathbf{z}) \ .$$

Now we are ready to proceed with the proof.

*Proof.* Let $\mathcal{U} = H_{\mathcal{F}^*}(\mathcal{C})$, $\mathbf{z} = (u_i, v_i)_{i=1}^l \in \mathcal{U}^l$, $\mathbb{I}_{a \neq b} = \mathbb{I}[a \neq b]$ and

$$A(\mathbf{z}) \ = \ \mathrm{Avg}_{i \neq j}\left[\mathbb{I}_{u_i \neq u_j} \cdot \frac{v_i + v_j}{\|u_i - u_j\|^2}\right] \ .$$

Furthermore, assume that $\mathcal{P} = \{P_1, \dots, P_l\} \sim \mathcal{D}_{\mathcal{C}}(P_1, \dots, P_l \mid \forall i \neq j \in [l] : P_i \neq P_j)$ is an independent and identical copy of $\tilde{P}$. Then, by Corollary 3 of Maurer & Pontil (2019) and the fact that the Gaussian complexity of any set $A \subset \mathbb{R}^l$ is at most $2\sqrt{\log(l)}$ times its Rademacher complexity, with probability at least $1 - \delta$ over the selection of $\tilde{\mathcal{P}}$,

$$\max_{f \in \mathcal{F}^*} \left[\mathbb{E}_{\mathcal{P}}\left[A(H_f(\mathcal{P}))\right] - A(H_f(\tilde{\mathcal{P}}))\right]$$

$$\leq 2\sqrt{2\pi \log(l)} \left(2M(A) + J(A)\right) \cdot \mathbb{E}[R(H_{\mathcal{F}^*}(\tilde{\mathcal{P}}))] + K(A) \cdot \sqrt{l \log(1/\delta)} \ . \tag{12}$$

Since $\tilde{P}_1, \dots, \tilde{P}_l$ are sampled such that $\tilde{P}_i \neq \tilde{P}_j$ (for all $i \neq j \in [l]$) and $\Delta(\mathcal{F}^*) > 0$, we have $\mathbb{I}[\mu_f(\tilde{P}_i) \neq \mu_f(\tilde{P}_j)] = 1$ for all $i \neq j$, and so

$$A((H_f(\tilde{\mathcal{P}})) \ = \ \mathrm{Avg}_{i \neq j}\left[\mathbb{I}[\mu_f(\tilde{P}_i) \neq \mu_f(P_j)] \cdot V_f(\tilde{P}_i, \tilde{P}_j)\right] \ = \ \mathrm{Avg}_{i \neq j}[V_f(\tilde{P}_i, \tilde{P}_j)]$$

Similarly, we have $A((H_f(\mathcal{P})) = \mathrm{Avg}_{i \neq j}[V_f(P_i, P_j)]$, and by the linearity of expectation, $\mathbb{E}_{\mathcal{P}}\left[A(H_f(\mathcal{P}))\right] = \mathbb{E}_{P_c \neq P_{c'}}\left[V_f(P_c, P_{c'})\right]$. By substituting these into (12), we obtain that for any

$f \in \mathcal{F}^*$, we have

$$
\begin{aligned}
\mathbb{E}_{P_c \neq P_{c'}} \left[ V_f(P_c, P_{c'}) \right] \leq \; & \mathrm{Avg}_{i \neq j} V_f(\tilde{P}_i, \tilde{P}_j) \\
& + 2\sqrt{2\pi \log(l)} \left( 2M(A) + J(A) \right) \cdot \mathbb{E}[R(H_{\mathcal{F}^*}(\tilde{\mathcal{P}}))] \\
& + K(A) \cdot \sqrt{l \log(1/\delta)} \; .
\end{aligned}
\tag{13}
$$

To finish the proof, we bound $M(A)$, $J(A)$ and $K(A)$. Let $j \in [l]$ be some index and $\mathbf{z}' = (z_i)_{i=1}^l = (u_i', v_i')_{i=1}^l \in \mathcal{U}^l$ be a vector, such that, $z_i' = z_i$ for all $i \neq j$ and $z_j' \neq z_j$. Then,

$$
\begin{aligned}
D_{z_j'}^j A(\mathbf{z}) \; &= \; A(\mathbf{z}) - A(\mathbf{z}') \\
&= \; \frac{1}{l(l-l)} \sum_{i \in [l]:\, i \neq j} \left[ \mathbb{I}_{u_i \neq u_j} \frac{v_i + v_j}{\|u_i - u_j\|^2} - \mathbb{I}_{u_i \neq u_j'} \frac{v_i + v_j'}{\|u_i - u_j'\|^2} \right] \; ,
\end{aligned}
\tag{14}
$$

since only pairs involving $j$ are non-zero in the average defining $A$. To simplify notation, let $a = \|u_i - u_j\|$ and $b = \|u_i - u_j'\|$ and $c = \|z_j - z_j'\| = \|(u_j, v_j) - (u_j', v_j')\|$. Next we bound the terms in the sum in (14) individually. We divide the analysis into three cases:

**Case 1.** Assume $u_i = u_j = u_j'$. In this case we simply have

$$
\left( \mathbb{I}_{u_i \neq u_j} \frac{v_i + v_j}{\|u_i - u_j\|^2} - \mathbb{I}_{u_i \neq u_j'} \frac{v_i + v_j'}{\|u_i - u_j'\|^2} \right) = 0 \; .
$$

**Case 2.** Assume $u_i \neq u_j$ and $u_i = u_j'$ (a bound can be obtained similarly for $u_i = u_j$ and $u_i \neq u_j'$). In this case, since $z_i, z_j \in \mathcal{U}$, we have $v_i, v_j \leq \sup_{f \in \mathcal{F}^*, P' \in \mathcal{C}} \mathrm{Var}_f(P')$, implying

$$
\left| \mathbb{I}_{u_i \neq u_j} \frac{v_i + v_j}{\|u_i - u_j\|^2} - \mathbb{I}_{u_i \neq u_j'} \frac{v_i + v_j'}{\|u_i - u_j'\|^2} \right| \leq \frac{v_i + v_j}{\|u_i - u_j\|^2} \leq \frac{2 \sup_{f \in \mathcal{F}^*, P' \in \mathcal{C}} \mathrm{Var}_f(P')}{\Delta(\mathcal{F}^*)^2} \; .
\tag{15}
$$

In addition, since $c \geq \|u_i - u_j\| \geq \Delta(\mathcal{F}^*)$, we have

$$
c^{-1} \left| \mathbb{I}_{u_i \neq u_j} \frac{v_i + v_j}{\|u_i - u_j\|^2} - \mathbb{I}_{u_i \neq u_j'} \frac{v_i + v_j'}{\|u_i - u_j'\|^2} \right| \leq \frac{2 \sup_{f \in \mathcal{F}^*, P' \in \mathcal{C}} \mathrm{Var}_f(P')}{\Delta(\mathcal{F}^*)^3} \; .
\tag{16}
$$

**Case 3.** Assume $u_i \neq u_j$ and $u_i \neq u_j'$. By simple algebraic manipulations, we have

$$
\begin{aligned}
& \left| \mathbb{I}_{u_i \neq u_j} \frac{v_i + v_j}{\|u_i - u_j\|^2} - \mathbb{I}_{u_i \neq u_j'} \frac{v_i + v_j'}{\|u_i - u_j'\|^2} \right| \\
&= \left| \frac{v_i + v_j}{\|u_i - u_j\|^2} - \frac{v_i + v_j'}{\|u_i - u_j'\|^2} \right| \\
&= \left| (v_i + v_j) \frac{(b - a)(b + a)}{a^2 b^2} + \frac{v_j - v_j'}{b^2} \right| \\
&\leq (v_i + v_j) \frac{|b - a| \cdot (b + a)}{a^2 b^2} + \frac{|v_j - v_j'|}{b^2} \\
&\leq (v_i + v_j) \frac{\|u_j - u_j'\| \cdot (b + a)}{a^2 b^2} + \frac{|v_j - v_j'|}{b^2} \; ,
\end{aligned}
$$

where the last inequality follows from the triangle inequality. Since $\|u_j - u'_j\| \leq \|u_j\| + \|u'_j\| \leq 2\sup_{x \in \mathcal{X}, \, f \in \mathcal{F}^*} \|f(x)\|$ and $v_i, v_j \leq \sup_{f \in \mathcal{F}^*, \, P' \in \mathcal{C}} \mathrm{Var}_f(P')$, we have

$$\left| \mathbb{I}_{u_i \neq u_j} \frac{v_i + v_j}{\|u_i - u_j\|^2} - \mathbb{I}_{u_i \neq u'_j} \frac{v_i + v'_j}{\|u_i - u'_j\|^2} \right|$$

$$\leq 4 \sup_{x \in \mathcal{X}, \, f \in \mathcal{F}^*} \|f(x)\| \cdot \sup_{f \in \mathcal{F}^*, \, P' \in \mathcal{C}} \mathrm{Var}_f(P') \cdot \left( \frac{1}{ab^2} + \frac{1}{a^2 b} \right) + \frac{2 \sup_{f \in \mathcal{F}^*, P' \in \mathcal{C}} \mathrm{Var}_f(P')}{\Delta(\mathcal{F}^*)^2}$$

$$\leq \frac{8 \sup_{x \in \mathcal{X}, \, f \in \mathcal{F}^*} \|f(x)\| \cdot \sup_{f \in \mathcal{F}^*, \, P' \in \mathcal{C}} \mathrm{Var}_f(P')}{\Delta(\mathcal{F}^*)^3} + \frac{2 \sup_{f \in \mathcal{F}^*, P' \in \mathcal{C}} \mathrm{Var}_f(P')}{\Delta(\mathcal{F}^*)^2}$$

We notice that $\|u_j - u'_j\| \leq c$ and $|v_j - v'_j| \leq c$. Therefore, we obtain

$$c^{-1} \left| \mathbb{I}_{u_i \neq u_j} \frac{v_i + v_j}{\|u_i - u_j\|^2} - \mathbb{I}_{u_i \neq u'_j} \frac{v_i + v'_j}{\|u_i - u'_j\|^2} \right|$$

$$\leq (v_i + v_j) \cdot \left( \frac{1}{ab^2} + \frac{1}{a^2 b} \right) + \frac{1}{b^2}$$

$$\leq \frac{4 \sup_{f \in \mathcal{F}^*, \, P' \in \mathcal{C}} \mathrm{Var}_f(P')}{\Delta(\mathcal{F}^*)^3} + \frac{1}{\Delta(\mathcal{F}^*)^2} \;.$$

Therefore, in each case we have

$$\left| \mathbb{I}_{u_i \neq u_j} \frac{v_i + v_j}{\|u_i - u_j\|^2} - \mathbb{I}_{u_i \neq u'_j} \frac{v_i + v'_j}{\|u_i - u'_j\|^2} \right|$$

$$\leq \frac{4 \sup_{x \in \mathcal{X}, \, f \in \mathcal{F}^*} \|f(x)\| \cdot \sup_{f \in \mathcal{F}^*, \, P' \in \mathcal{C}} \mathrm{Var}_f(P')}{\Delta(\mathcal{F}^*)^3} + \frac{2 \sup_{f \in \mathcal{F}^*, P' \in \mathcal{C}} \mathrm{Var}_f(P')}{\Delta(\mathcal{F}^*)^2}$$

and also

$$c^{-1} \left| \mathbb{I}_{u_i \neq u_j} \frac{v_i + v_j}{\|u_i - u_j\|^2} - \mathbb{I}_{u_i \neq u'_j} \frac{v_i + v'_j}{\|u_i - u'_j\|^2} \right| \leq \frac{4 \sup_{f \in \mathcal{F}^*, P' \in \mathcal{C}} \mathrm{Var}_f(P')}{\Delta(\mathcal{F}^*)^3} + \frac{1}{\Delta(\mathcal{F}^*)^2} \;.$$

Hence,

$$\left| D_{z'_j}^j A(\mathbf{z}) \right| \leq \frac{8 \sup_{x \in \mathcal{X}, \, f \in \mathcal{F}^*} \|f(x)\| \cdot \sup_{f \in \mathcal{F}^*, \, P' \in \mathcal{C}} \mathrm{Var}_f(P')}{l \cdot \Delta(\mathcal{F}^*)^3} + \frac{2 \sup_{f \in \mathcal{F}^*, P' \in \mathcal{C}} \mathrm{Var}_f(P')}{l \cdot \Delta(\mathcal{F}^*)^2} \;, \quad (17)$$

and

$$\frac{\left| D_{z'_j}^j A(\mathbf{z}) \right|}{\|z_j - z'_j\|} \leq \frac{4 \sup_{f \in \mathcal{F}^*, \, P' \in \mathcal{C}} \mathrm{Var}_f(P')}{l \cdot \Delta(\mathcal{F}^*)^3} + \frac{1}{l \cdot \Delta(\mathcal{F}^*)^2} \;. \quad (18)$$

Therefore, $K(A)$ and $M(A)$ are also bounded by the right hand sides of (17) and (18). Next, we upper bound $J(A)$. Let $r \neq j$ and $z'_r \in \mathcal{U}$. We have

$$\frac{D_{z'_r}^r D_{z'_j}^j A(\mathbf{z})}{\|z_j - z'_j\|} = \frac{1}{l(l-1)} \cdot \frac{\mathbb{I}_{u_r \neq u_j} \frac{v_r + v_j}{\|u_r - u_j\|^2} - \mathbb{I}_{u_r \neq u'_j} \frac{v_r + v'_j}{\|u_r - u'_j\|^2}}{c}$$

$$- \frac{1}{l(l-1)} \cdot \frac{\mathbb{I}_{u'_r \neq u_j} \frac{v'_r + v_j}{\|u'_r - u_j\|^2} - \mathbb{I}_{u'_r \neq u'_j} \frac{v'_r + v'_j}{\|u'_r - u'_j\|^2}}{c}$$

$$\leq \frac{1}{l(l-1)} \cdot \left| \frac{\mathbb{I}_{u_r \neq u_j} \frac{v_r + v_j}{\|u_r - u_j\|^2} - \mathbb{I}_{u_r \neq u'_j} \frac{v_r + v'_j}{\|u_r - u'_j\|^2}}{c} \right|$$

$$+ \frac{1}{l(l-1)} \cdot \left| \frac{\mathbb{I}_{u'_r \neq u_j} \frac{v'_r + v_j}{\|u'_r - u_j\|^2} - \mathbb{I}_{u'_r \neq u'_j} \frac{v'_r + v'_j}{\|u'_r - u'_j\|^2}}{c} \right| \;.$$

Hence, by (18) we have

$$J(A) \;\le\; \frac{1}{l-1} \left( \frac{8 \sup\limits_{f \in \mathcal{F}^*, P' \in \mathcal{C}} \mathrm{Var}_f(P')}{\Delta(\mathcal{F}^*)^3} + \frac{2}{\Delta(\mathcal{F}^*)^2} \right).$$

Substituting the bounds of $M(A)$, $K(A)$ and $J(A)$ into (13) proves the proposition. $\qquad\square$

## E  ANALYSIS FOR SECTIONS 4.1-4.2

In this section we analyze the asymptotic behaviour of $\epsilon_1^c(\delta)$ and $\epsilon_2^c(\delta)$ (see Section 4.1), as well as of $\mathbb{E}_{\tilde{\mathcal{P}}}[R(H_{\mathcal{F}^*}(\tilde{\mathcal{P}}))]$ (see Proposition 2) for ReLU neural networks. A ReLU neural feature map[2] is a function of the form $f(x) = W^q \sigma(W^{q-1} \ldots \sigma(W^1 x)) : \mathbb{R}^d \to \mathbb{R}^p$, where $\sigma$ is the element-wise ReLU function $\sigma(x) = \max(0, x)$, and $W^i \in \mathbb{R}^{d_{i+1} \times d_i}$ for $i \in [q]$, where $d_1 = d$ and $d_{q+1} = p$. Throughout this section, we use $\mathcal{F}$ to denote the set of ReLU neural feature maps (with the depth $q$ and the dimensions $d_1, \ldots, d_{q+1}$ of the layers fixed). The spectral complexity of a network $f$ is defined as $\mathcal{C}(f) = \max_{j \in [p]} \|W_j^q\| \cdot \prod_{r=1}^{q-1} \|W^r\|$ where for vectors, $\|\cdot\|$ denotes the Euclidean norm, while for matrices, it is the spectral ($L_2$-induced) norm. This quantity upper bounds the Lipschitz constant of $f$ and is similar in fashion to other (slightly different) notions of spectral complexity for neural networks (Golowich et al., 2017; Bartlett et al., 2017).

Throughout the section, for a given function $g : A \to B^k$ and $j \in [k]$, we denote the $j$'th coordinate of $g(x)$ by $g(x)_j$ and for a class $\mathcal{G} \subset \{g : A \to B^k\}$, we define $\mathcal{G}_{|j} = \{g(\cdot)_j : g \in \mathcal{G}\}$.

Before presenting the main claims of this section, following Poggio et al. (2020), we start with describing a canonical representation of ReLU neural networks (the proofs of our statements are given for this representation). Since $\max(0, ax) = a \max(0, x)$, for all $x \in \mathbb{R}$ and $a \ge 0$, any neural network $f = W^q \sigma(W^{q-1} \ldots \sigma(W^1 x)) \in \mathcal{F}$ can be represented as a modified network $f'(x) = V^q \sigma(V^{q-1} \ldots \sigma(V^1 x))$, where $V^q = W^q \cdot \prod_{r=1}^{q-1} \|W^r\|$ and $\forall r \le q - 1 : V^r = \frac{W^r}{\|W^r\|}$. In particular, $\|V^r\| = 1$ for all $r \le q - 1$ and, since $V^q = W_j^q \cdot \prod_{r=1}^{q-1} \|W^r\|$, $\max_{j \in p} \|V_j^q\| = \max_{j \in p} \|W_j^q\| \cdot \prod_{r=1}^{q-1} \|W^r\| = \mathcal{C}(f)$. Thus, for any $f \in \mathcal{F}$, there exists an equivalent neural network $f'$ which belongs to the set

$$\mathcal{F}^M = \left\{ W^q \sigma(W^{q-1} \ldots \sigma(W^1 x)) \in \mathcal{F} \mid \forall r \le q - 1 : \|W^r\| = 1 \text{ and } \max_{j \in [p]} \|W_j^q\| \le M \right\}$$

for any $M \ge \mathcal{C}(f)$. Therefore, $\mathcal{F} = \bigcup_{M=1}^{\infty} \mathcal{F}^M$ ($\mathcal{F}^M \subset \mathcal{F}$ is trivial by definition).

Next we present the first claim of the section. The following proposition shows that the first and second moments of ReLU neural feature maps with bounded spectral norms concentrate around their means. In particular, if a learning algorithm (in Section 4.1) is guaranteed to return a neural network $f$ with $\mathcal{C}(f) \le M$, then $\epsilon_1^c(\delta)$ and $\epsilon_2^c(\delta)$ are bounded by the right hand sides of (19) and (20), respectively. Note that both of these terms scale as $\mathcal{O}(\sqrt{\log(1/\delta)/m_c})$. The analysis is based on Theorem 1 of Golowich et al. (2017) and the proof of Theorem 1.1 of Bartlett et al. (2017).

**Proposition 3.** *Let $\tilde{\mathcal{P}} = \{\tilde{P}_c\}_{c=1}^l$ be a set of class-conditional distributions over a bounded set $\mathcal{X} \subset \mathbb{R}^d$ and let $m_1, \ldots, m_l \in \mathbb{N}$. Let $\mathcal{F}$ be the class of ReLU neural feature maps as defined above. Then, with probability at least $1 - \delta$ over the selection of $\tilde{S}_1 \sim \tilde{P}_1^{m_1}, \ldots, \tilde{S}_l \sim \tilde{P}_l^{m_l}$, for all $c \in [l]$ and $f \in \mathcal{F}$, we have*

$$\left\| \mathbb{E}_{\tilde{x} \sim \tilde{P}_c}[f(\tilde{x})] - \mathrm{Avg}_{\tilde{x} \in \tilde{S}_c}[f(\tilde{x})] \right\|$$
$$\le \frac{p(\mathcal{C}(f) + 1) \sup_{x \in \mathcal{X}} \|x\|}{\sqrt{m_c}} \left( 3\sqrt{q} + 2 + \sqrt{\frac{\log(4pl/\delta)}{2}} + \sqrt{\log(\mathcal{C}(f) + 1)} \right) \qquad (19)$$

---

[2]ReLU neural networks for classification typically have an additional soft-max layer on top their feature map.

*and*

$$\left| \mathbb{E}_{\tilde{x} \sim \tilde{P}_c} \left[ \|f(\tilde{x})\|^2 \right] - \mathrm{Avg}_{\tilde{x} \in \tilde{S}_c} \left[ \|f(\tilde{x})\|^2 \right] \right|$$
$$\leq \frac{pM^2 \sup_{x \in \mathcal{X}} \|x\|^2}{\sqrt{m_c}} \left( 6\sqrt{q} + 4 + 3\sqrt{\frac{\log(4l/\delta)}{2}} + 3\sqrt{\log(\mathcal{C}(f)+1)} \right) . \tag{20}$$

*Proof.* We prove that, for any fixed $c \in [l]$ and $M \in \mathbb{N}$, (19) and, respectively, (20) hold for all $f \in \mathcal{F}^M$ simultaneously with probability at least $1 - \frac{\delta}{2lM(M+1)}$. Then taking the union bound over $c$ and $M$ proves the proposition (since $\mathcal{F} = \bigcup_{M=1}^{\infty} \mathcal{F}^M$).

Fix $c \in [l]$ and $M \in \mathbb{N}$. By the triangle inequality, for any $f \in \mathcal{F}$, we have

$$\left\| \mathbb{E}_{\tilde{x} \sim \tilde{P}_c} [f(\tilde{x})] - \mathrm{Avg}_{\tilde{x} \in \tilde{S}_c} [f(\tilde{x})] \right\| \leq \sum_{j=1}^{p} \left| \mathbb{E}_{\tilde{x} \sim \tilde{P}_c} [f(\tilde{x})_j] - \mathrm{Avg}_{\tilde{x} \in \tilde{S}_c} [f(\tilde{x})_j] \right| . \tag{21}$$

By Theorem 3.3 of Mohri et al. (2018), with probability at least $1 - \frac{\delta}{2plM(M+1)}$ over the selection of $\tilde{S}_c \sim \tilde{P}_c^{m_c}$, for any $f \in \mathcal{F}^M$, we have

$$\left| \mathbb{E}_{\tilde{x} \sim \tilde{P}_c} [f(\tilde{x})_j] - \mathrm{Avg}_{\tilde{x} \in \tilde{S}_c} [f(\tilde{x})_j] \right|$$
$$\leq \frac{2R(\mathcal{F}_{|j}^M(\tilde{S}_c))}{m_c} + 3 \sup_{x \in \mathcal{X}, \, f' \in \mathcal{F}^M} |f'(x)_j| \cdot \sqrt{\frac{\log(4plM(M+1)/\delta)}{2m_c}} . \tag{22}$$

The first term on the right hand side can be bounded using Theorem 1 of Golowich et al. (2017),[3] stating that

$$R(\mathcal{F}_{|j}^M(\tilde{S}_c)) \leq \sqrt{m_c} \left( \sqrt{2\log(2)q} + 1 \right) M \sup_{x \in \mathcal{X}} \|x\| \leq \sqrt{m_c} \left( 1.5\sqrt{q} + 1 \right) M \sup_{x \in \mathcal{X}} \|x\| . \tag{23}$$

Moreover, for any $f' \in \mathcal{F}^M$ we have $|f'(x)_j| \leq M \cdot \sup_{x \in \mathcal{X}} \|x\|$. Substituting these inequalities into (22) implies

$$\left| \mathbb{E}_{\tilde{x} \sim \tilde{P}_c} [f(\tilde{x})_j] - \mathrm{Avg}_{\tilde{x} \in \tilde{S}_c} [f(\tilde{x})_j] \right|$$
$$\leq \frac{(3\sqrt{q} + 2)M \cdot \sup_{x \in \mathcal{X}} \|x\|}{\sqrt{m_c}} + 3M \cdot \sup_{x \in \mathcal{X}} \|x\| \cdot \sqrt{\frac{\log(4plM(M+1)/\delta)}{2m_c}} . \tag{24}$$

By the union bound, (24) holds for all $j \in [p]$ simultaneously with probability at least $1 - \frac{\delta}{2lM(M+1)}$. Combining this with (21), we obtain that for any fixed $c \in [l]$ and $M \in \mathbb{N}$, with probability at least $1 - \frac{\delta}{2lM(M+1)}$, for all $f \in \mathcal{F}^M \setminus \mathcal{F}^{M-1}$ we have

$$\left\| \mathbb{E}_{\tilde{x} \sim \tilde{P}_c} [f(\tilde{x})] - \mathrm{Avg}_{\tilde{x} \in \tilde{S}_c} [f(\tilde{x})] \right\|$$
$$\leq \frac{pM \sup_{x \in \mathcal{X}} \|x\|}{\sqrt{m_c}} \left( 3\sqrt{q} + 2 + 3\sqrt{\frac{\log(4plM(M+1)/\delta)}{2}} \right)$$
$$\leq \frac{pM \sup_{x \in \mathcal{X}} \|x\|}{\sqrt{m_c}} \left( 3\sqrt{q} + 2 + 3\sqrt{\frac{\log(4pl/\delta)}{2}} + 3\sqrt{\log(M+1)} \right)$$
$$\leq \frac{p(\mathcal{C}(f)+1) \sup_{x \in \mathcal{X}} \|x\|}{\sqrt{m_c}} \left( 3\sqrt{q} + 2 + 3\sqrt{\frac{\log(4pl/\delta)}{2}} + 3\sqrt{\log(\mathcal{C}(f)+2)} \right) , \tag{25}$$

---

[3] The original statement makes use of the Frobenius norms of the matrices, however, the proof is exactly the same if we replace the Frobenius norm with the spectral norm.

where the last inequality follows from the fact that $M \leq \mathcal{C}(f) + 1$, since $\mathcal{C}(f) \in [M - 1, M]$. Next, we prove the second inequality for fixed $c$ and $M$. As in (22), by Theorem 3.3 of Mohri et al. (2018), with probability at least $1 - \frac{\delta}{2lM(M+1)}$ over the selection of $\tilde{S}_c$, for all $f \in \mathcal{F}^M$, we have

$$
\left| \mathbb{E}_{\tilde{x} \sim \tilde{P}_c} \left[ \|f(\tilde{x})\|^2 \right] - \mathrm{Avg}_{\tilde{x} \in \tilde{S}_c} \left[ \|f(\tilde{x})\|^2 \right] \right|
$$
$$
\leq \frac{2R(\mathcal{G}^s(\tilde{S}_c))}{m_c} + 3 \sup_{x \in \mathcal{X}, \, f \in \mathcal{F}^M} \|f(x)\|^2 \cdot \sqrt{\frac{\log(4lM(M+1)/\delta)}{2m_c}} \,, \tag{26}
$$

where $\mathcal{G}^s = \{ \|f(\cdot)\|^2 \mid f \in \mathcal{F}^M \}$. By definition, with $\epsilon = (\epsilon_1, \ldots, \epsilon_{m_c})$ denoting i.i.d. Rademacher random variables (i.e., random variables taking values $\pm 1$ with probability $1/2$ each),

$$
R(\mathcal{G}^s(\tilde{S}_c)) = \mathbb{E}_\epsilon \left[ \sup_{f \in \mathcal{F}^M} \left( \sum_{i=1}^{m_c} \epsilon_i \|f(\tilde{x}_{ci})\|^2 \right) \right] = \mathbb{E}_\epsilon \left[ \sup_{f \in \mathcal{F}^M} \left( \sum_{i=1}^{m_c} \sum_{j=1}^{p} \epsilon_i f(\tilde{x}_{ci})_j^2 \right) \right]
$$
$$
\leq \sum_{j=1}^{p} \mathbb{E}_\epsilon \left[ \sup_{f \in \mathcal{F}^M} \left( \sum_{i=1}^{m_c} \epsilon_i f(\tilde{x}_{ci})_j^2 \right) \right] . \tag{27}
$$

Note that the $j$th terms (in the square bracket) at the right-hand side above is the Rademacher complexity of the composite function class $g \circ \mathcal{F}_j^M = \{ g \circ f' | f' \in \mathcal{F}_j^M \}$ for $g(y) = y^2$. Since $\sup_{x \in \mathcal{X}} |f'(x)_j| \leq M \sup_{x \in \mathcal{X}} \|x\|$ and $g$ is $2M \sup_{x \in \mathcal{X}} \|x\|$-Lipschitz on $[-M \sup_{x \in \mathcal{X}} \|x\|, M \sup_{x \in \mathcal{X}} \|x\|]$, by Lemma 1.1 in Lecture 17 of Kakade & Tewari (2008), the Rademacher complexity of this composite function class can be bounded as

$$
R(g \circ \mathcal{F}_j^M(\tilde{S}_c)) \leq 2M \sup_{x \in \mathcal{X}} \|x\| \cdot R(\mathcal{F}_j^M(\tilde{S}_c)). \tag{28}
$$

Combining with (27) and (23) yields

$$
R(\mathcal{G}^s(\tilde{S}_c)) \leq \sum_{j=1}^{p} R(g \circ \mathcal{F}_j^M) \leq 2M \sup_{x \in \mathcal{X}} \|x\| \sum_{j=1}^{p} R(\mathcal{F}_j^M) \leq pM^2 \sup_{x \in \mathcal{X}} \|x\|^2 \sqrt{m_c} \, (3\sqrt{q} + 2) \,. \tag{29}
$$

Substituting into (26) and using $\sup_{x \in \mathcal{X}, \, f' \in \mathcal{F}^M} \|f'(x)\|^2 \leq pM^2 \sup_{x \in \mathcal{X}} \|x\|^2$ imply, similarly to (25), that with probability at least $1 - \frac{\delta}{2lM(M+1)}$ over the selection of $\tilde{S}_c$, for all $f \in \mathcal{F}^M \setminus \mathcal{F}^{M-1}$ simultaneously we have

$$
\left| \mathbb{E}_{\tilde{x} \sim \tilde{P}_c} \left[ \|f(\tilde{x})\|^2 \right] - \mathrm{Avg}_{\tilde{x} \in \tilde{S}_c} \left[ \|f(\tilde{x})\|^2 \right] \right|
$$
$$
\leq \frac{pM^2 \sup_{x \in \mathcal{X}} \|x\|^2}{\sqrt{m_c}} \left( 6\sqrt{q} + 4 + 3\sqrt{\frac{\log(4l/\delta)}{2}} + 3\sqrt{\log(\mathcal{C}(f) + 2)} \right) ,
$$

finishing the proof. $\qquad \square$

Next we show that the Rademacher complexity in Proposition 2 scales as $\mathcal{O}(\sqrt{l})$ for ReLU neural networks with bounded spectral complexities.

**Proposition 4.** *Let $\mathcal{F}^* = \{ f \in \mathcal{F} | \mathcal{C}(f) \leq M \}$ be a class of ReLU neural networks with. Then,*

$$
\mathbb{E}_{\tilde{\mathcal{P}}}[R(H_{\mathcal{F}^*}(\tilde{\mathcal{P}}))] \leq \sqrt{l}(1.5\sqrt{q} + 1)M \sup_{x \in \mathcal{X}} \|x\| \left( 1 + 4pM \sup_{x \in \mathcal{X}} \|x\| \right)
$$

*Proof.* As discussed at the beginning of the section, $\mathcal{F}^*$ and $\mathcal{F}^M$ define the same function class through different representations. Thus it suffices to consider $\mathcal{F}^M$ in place of $\mathcal{F}^*$ (as trivially $\mathbb{E}_{\tilde{\mathcal{P}}}[R(H_{\mathcal{F}^*}(\tilde{\mathcal{P}}))] = \mathbb{E}_{\tilde{\mathcal{P}}}[R(H_{\mathcal{F}^M}(\tilde{\mathcal{P}}))]$).

Fix $\tilde{P} = (\tilde{P}_1, \ldots, \tilde{P}_l)$. With $\epsilon = (\epsilon_{cj})_{c \in [l], j \in \{0\} \cup [p]}$ denoting i.i.d. Rademacher random variables, by the definition of $H_{\mathcal{F}^M}$ and because $\sup_{a \in A} [f(a) + g(a)] \leq \sup_{a \in A} f(a) + \sup_{a \in A} g(a)$, we

have

$$
R(H_{\mathcal{F}^M}(\tilde{\mathcal{P}})) = \mathbb{E}_\epsilon \left[ \sup_{f \in \mathcal{F}^M} \left( \sum_{i=1}^l \sum_{j=1}^p \epsilon_{cj} \mu_f(\tilde{P}_c)_j + \epsilon_{c0} \mathrm{Var}_f(\tilde{P}_c) \right) \right]
$$

$$
\leq \sum_{j=1}^p \mathbb{E}_\epsilon \left[ \sup_{f \in \mathcal{F}^M} \left( \sum_{c=1}^l \epsilon_{cj} \mu_f(\tilde{P}_c)_j \right) \right] + \mathbb{E}_\epsilon \left[ \sup_{f \in \mathcal{F}^M} \left( \sum_{c=1}^l \epsilon_{c0} \mathrm{Var}_f(\tilde{P}_c) \right) \right] . \quad (30)
$$

We bound the two terms in (30) separately. The terms in the first summation can be bounded as

$$
\sup_{f \in \mathcal{F}^M} \left( \sum_{c=1}^l \epsilon_{cj} \mu_f(\tilde{P}_c)_j \right) = \sup_{f \in \mathcal{F}^M} \left( \sum_{c=1}^l \epsilon_{cj} \mathbb{E}_{\tilde{x}_c \sim \tilde{P}_c}[f(\tilde{x}_c)_j] \right)
$$

$$
\leq \mathbb{E}_{\tilde{x}_1 \sim \tilde{P}_1, \ldots, \tilde{x}_l \sim \tilde{P}_l} \mathbb{E}_\epsilon \left[ \sup_{f \in \mathcal{F}^M} \left( \sum_{c=1}^l \epsilon_{cj} f(\tilde{x}_c)_j \right) \right] \quad (31)
$$

$$
= \mathbb{E}_{\tilde{x}_1 \sim \tilde{P}_1, \ldots, \tilde{x}_l \sim \tilde{P}_l} R \left( \mathcal{F}_{|j}^M(\{\tilde{x}_c\}_{c=1}^l) \right)
$$

$$
\leq \sqrt{l}(1.5\sqrt{q}+1) M \sup_{x \in \mathcal{X}} \|x\| ,
$$

where the last inequality holds because of (23) (with $l$ instead of $m_c$ samples). The elements of the second summation (30) can be bounded as

$$
\mathbb{E}_\epsilon \left[ \sup_{f \in \mathcal{F}^M} \left( \sum_{c=1}^l \epsilon_{c0} \mathrm{Var}_f(\tilde{P}_c) \right) \right]
$$

$$
= \mathbb{E}_\epsilon \left[ \sup_{f \in \mathcal{F}^M} \left( \sum_{c=1}^l \epsilon_{c0} \left( \mathbb{E}_{\tilde{x}_c \sim \tilde{P}_c}[\|f(\tilde{x}_c)\|^2] - \|\mathbb{E}_{\tilde{x}_c \sim \tilde{P}_c}[f(\tilde{x}_c)]\|^2 \right) \right) \right]
$$

$$
\leq \mathbb{E}_\epsilon \left[ \sup_{f \in \mathcal{F}^M} \left( \sum_{c=1}^l \epsilon_{c0} \mathbb{E}_{\tilde{x}_c \sim \tilde{P}_c}[\|f(\tilde{x}_c)\|^2] \right) \right] + \mathbb{E}_\epsilon \left[ \sup_{f \in \mathcal{F}^M} \left( -\sum_{c=1}^l \epsilon_{c0} \|\mathbb{E}_{\tilde{x}_c \sim \tilde{P}_c}[f(\tilde{x}_c)]\|^2 \right) \right]
$$

$$
= \mathbb{E}_\epsilon \left[ \sup_{f \in \mathcal{F}^M} \left( \sum_{c=1}^l \epsilon_{c0} \mathbb{E}_{\tilde{x}_c \sim \tilde{P}_c}[\|f(\tilde{x}_c)\|^2] \right) \right] + \mathbb{E}_\epsilon \left[ \sup_{f \in \mathcal{F}^M} \left( \sum_{c=1}^l \epsilon_{c0} \|\mathbb{E}_{\tilde{x}_c \sim \tilde{P}_c}[f(\tilde{x}_c)]\|^2 \right) \right] , \quad (32)
$$

where the last equation follows from the fact that $\epsilon_{c0}$ are distributed symmetrically around $0$. We can bound the first term in (32) above using (29) (again, with $l$ instead of $m_c$ samples) as

$$
\mathbb{E}_\epsilon \left[ \sup_{f \in \mathcal{F}^M} \left( \sum_{c=1}^l \epsilon_{c0} \mathbb{E}_{\tilde{x}_c \sim \tilde{P}_c}[\|f(\tilde{x}_c)\|^2] \right) \right]
$$

$$
\leq \mathbb{E}_{\tilde{x}_1 \sim \tilde{P}_1, \ldots, \tilde{x}_l \sim \tilde{P}_l} \mathbb{E}_\epsilon \left[ \sup_{f \in \mathcal{F}^M} \left( \sum_{c=1}^l \epsilon_{c0} \|f(\tilde{x}_c)\|^2 \right) \right] \leq pM^2 \sup_{x \in \mathcal{X}} \|x\|^2 \sqrt{l} \left( 3\sqrt{q}+2 \right) \quad (33)
$$

The second term in (32) can be bounded similarly, but because now the expectation is inside the squared norm, we need to provide a few more steps:

$$
\mathbb{E}_\epsilon \left[ \sup_{f \in \mathcal{F}^M} \left( \sum_{c=1}^l \epsilon_{c0} \|\mathbb{E}_{\tilde{x}_c \sim \tilde{P}_c}[f(\tilde{x}_c)]\|^2 \right) \right]
$$

$$
= \mathbb{E}_\epsilon \left[ \sup_{f \in \mathcal{F}^M} \left( \sum_{c=1}^l \sum_{j=1}^p \epsilon_{c0} \left( \mathbb{E}_{\tilde{x}_c \sim \tilde{P}_c}[f(\tilde{x}_c)_j] \right)^2 \right) \right] \quad (34)
$$

$$
\leq \sum_{j=1}^p \mathbb{E}_\epsilon \left[ \sup_{f \in \mathcal{F}^M} \left( \sum_{c=1}^l \epsilon_{c0} \left( \mathbb{E}_{\tilde{x}_c \sim \tilde{P}_i}[f(\tilde{x}_c)_j] \right)^2 \right) \right] .
$$

Now similarly to (28),

$$\max_{j \in [p]} \sup_{\tilde{P}} |\mathbb{E}_{\tilde{x} \sim \tilde{P}}[f(\tilde{x})_j]| \leq \max_{j \in [p]} \sup_{\tilde{P}} \mathbb{E}_{\tilde{x} \sim \tilde{P}}[|f(\tilde{x})_j|] \leq M \cdot \sup_{x \in \mathcal{X}} \|x\|$$

implies (via Lemma 1.1 in Lecture 17 of Kakade & Tewari (2008))

$$
\begin{aligned}
&\mathbb{E}_\epsilon \left[ \sup_{f \in \mathcal{F}^M} \left( \sum_{c=1}^{l} \epsilon_{c0} \left( \mathbb{E}_{\tilde{x}_c \sim \tilde{P}_c}[f(\tilde{x}_c)_j] \right)^2 \right) \right] \\
&\leq 2M \sup_{x \in \mathcal{X}} \|x\| \cdot \mathbb{E}_\epsilon \left[ \sup_{f \in \mathcal{F}^M} \left( \sum_{c=1}^{l} \epsilon_{c0} \mathbb{E}_{\tilde{x}_c \sim \tilde{P}_c}[f(\tilde{x}_c)_j] \right) \right] \\
&\leq 2M \sup_{x \in \mathcal{X}} \|x\| \cdot \mathbb{E}_{\tilde{x}_c \sim \tilde{P}_c} \mathbb{E}_\epsilon \left[ \sup_{f \in \mathcal{F}^M} \left( \sum_{c=1}^{l} \epsilon_{c0} f(\tilde{x}_c)_j \right) \right] \\
&= 2M \sup_{x \in \mathcal{X}} \|x\| \cdot \mathbb{E}_{\tilde{P}} \mathbb{E}_{\tilde{x}_c \sim \tilde{P}_c} [R(\mathcal{F}_{|j}^M(\{\tilde{x}_c\}_{c=1}^{l}))] \\
&\leq M^2 \sup_{x \in \mathcal{X}} \|x\|^2 \sqrt{l} \left( 3\sqrt{q} + 2 \right),
\end{aligned}
\tag{35}
$$

where in the last inequality we used again Theorem 1 of Golowich et al. (2017) adapted to the spectral norm, as in (23). Combining (34) and (35) gives

$$\mathbb{E}_\epsilon \left[ \sup_{f \in \mathcal{F}^M} \left( \sum_{c=1}^{l} \epsilon_{c0} \|\mathbb{E}_{\tilde{x}_c \sim \tilde{P}_c}[f(\tilde{x}_c)]\|^2 \right) \right] \leq pM^2 \sup_{x \in \mathcal{X}} \|x\|^2 \sqrt{l} \left( 3\sqrt{q} + 2 \right)$$

Substituting this and (33) into (32) gives

$$\mathbb{E}_\epsilon \left[ \sup_{f \in \mathcal{F}^M} \left( \sum_{c=1}^{l} \epsilon_{c0} \mathrm{Var}_f(\tilde{P}_c) \right) \right] \leq 2pM^2 \sup_{x \in \mathcal{X}} \|x\|^2 \sqrt{l} \left( 3\sqrt{q} + 2 \right).$$

Plugging in this and (31) into (30) gives

$$R(H_{\mathcal{F}^M}(\tilde{\mathcal{P}})) \leq \sqrt{l}(1.5\sqrt{q} + 1)M \sup_{x \in \mathcal{X}} \|x\| \left( 1 + 4pM \sup_{x \in \mathcal{X}} \|x\| \right),$$

giving the desired bound. $\qquad \square$

## F    ANALYSIS FOR SECTION 4.2

**Lemma 2.** *Let $X_1, \ldots, X_n$ be a set of i.i.d. uniformly distributed points in the $p$-dimensional unit cube. There is a positive constant $C > 0$ (independent of $n$ and $p$), such that, $\mathbb{E}\left[\min_{i \neq j \in [n]} \|X_i - X_j\|\right] \geq C \cdot n^{-2/p} \sqrt{p}$.*

*Proof.* For any pair $i \neq j \in [n]$,

$$\Pr[\|X_i - X_j\| \geq D] \geq 1 - \frac{\pi^{p/2}}{\Gamma(p/2 + 1)} D^p,$$

since the volume of the $p$-dimensional ball of radius $D$ is $\frac{\pi^{p/2}}{\Gamma(p/2+1)} D^p$ and the volume of the unit cube is 1. Hence, by the union bound over all pairs $i \neq j \in [n]$:

$$\Pr\left[\min_{i \neq j} \|X_i - X_j\| \geq D\right] \geq \max\left(0, 1 - \frac{n(n-1)}{2} \frac{\pi^{p/2}}{\Gamma(p/2+1)} D^p\right) = h(n, D).$$

Since $h(n, D) > 0$ if and only if $D < D^* = M^{-1/p}$, where $M = \frac{n(n-1)}{2} \frac{\pi^{p/2}}{\Gamma(p/2+1)}$, we obtain

$$
\begin{aligned}
\mathbb{E}\left[\min_{i \neq j} \|X_i - X_j\|\right] &= \int_0^\infty \Pr\left[\min_{i \neq j} \|X_i - X_j\| \geq D\right] dD \\
&\geq \int_0^{D^*} h(l, D) \, dD = D^* - M\frac{(D^*)^{p+1}}{(p+1)} \\
&= D^*\left(1 - M\frac{(D^*)^p}{p+1}\right) \\
&= D^*\left(1 - \frac{1}{p+1}\right) \\
&= \left(\frac{2}{n(n-1)}\right)^{1/p} \cdot \frac{\Gamma(p/2+1)^{1/p}}{\pi^{1/2}} \geq C \cdot n^{-2/p}\sqrt{p},
\end{aligned}
$$

where the last inequality follows from Ramanujan's approximations of the Gamma function (Karatsuba, 2001) for some $C > 0$ independent of $n, p$. $\qquad\square$

## G ANALYSIS FOR SECTION 4.3

In the following proposition we provide a formal statement of the analysis provided in Section 4.3. We consider a balanced $k$-class classification problem and a given feature map $f$ (e.g., pretrained). We upper bound the classification error of the nearest-mean classifier $h_{f,S}(x) = \arg\min_{c \in [k]} \|f(x) - \mu_f(S_c)\|$ in terms of the averaged CDNV, that is, $\text{Avg}_{i \neq j}[V_f(P_i, P_j)]$. Therefore, if the averaged CDNV is small, then, we expect the nearest-mean classifier to have a small classification error.

**Proposition 5.** *Let $T$ be a balanced $k$-class classification problem with distribution $P$ along with class-conditional distributions $\mathcal{P} = \{P_c\}_{c=1}^k$. Let $S = \bigcup_{c=1}^k S_c$ be a dataset, such that $S_c \sim P_c^{n_c}$ ($n_1 = \cdots = n_k = n_c \in \mathbb{N}$). Let $f : \mathbb{R}^d \to \mathbb{R}^p$ be any feature map and let $h_{f,S}(x) = \arg\min_{c \in [k]} \|f(x) - \mu_f(S_c)\|$ be the nearest-mean classifier. Then, we have*

$$
\mathbb{E}[\text{Err}] := \mathbb{E}_S \mathbb{E}_{(x,y) \sim P} \mathbb{I}[h(x) \neq y] \leq 16(k-1)\left[\frac{1}{s(f,\mathcal{P})} + \frac{1}{n_c}\right] \text{Avg}_{i \neq j}[V_f(P_i, P_j)],
$$

*where $s(f, \mathcal{P}) = p$ if $\{f \circ P_c\}_{c=1}^k$ are spherically symmetric and $s(f, \mathcal{P}) = 1$ otherwise.*

*Proof.* First note that

$$
\begin{aligned}
\mathbb{E}[\text{Err}] &= \mathbb{E}_S \mathbb{E}_{(x,y) \sim P} \mathbb{I}[h_{f,S}(x) \neq y] \\
&= \frac{1}{k}\sum_{i=1}^k \Pr[h_{f,S}(x_i) \neq i] \\
&= \frac{1}{k}\sum_{i \neq j} \Pr[h_{f,S}(x_i) = j].
\end{aligned}
\tag{36}
$$

In the following we will fix $i$ and $j \neq i$ and bound $\Pr[h_{f,S}(x_i) = j]$, which is the probability that $x_i \sim P_i$ is (wrongly) classified as $j \neq i$, for all three cases (general, symmetric, Gaussian). Let $c$ be either $i$ or $j$. Let $\mu_c := \mu_f(P_c)$ and $\hat{\mu}_c := \mu_f(S_c)$ and $u_i = f(x_i)$. Let $X_c := \|\hat{\mu}_c - u_i\|$ and $Y_c := \|u_i - \mu_c\|$ and $Z_c := \|\mu_c - \hat{\mu}_c\|$ for $c = i, j$. Let $\alpha_{ij} = \|\mu_i - \mu_j\|$ (or $\alpha$ for short since $i$ and $j$ are fixed) and $\sigma_c^2 := \text{Var}_f(P_c) \equiv \mathbb{E}[Y_c^2] = n_c \mathbb{E}[Z_c^2]$. Note that $V_f(P_i, P_j) = (\sigma_i^2 + \sigma_j^2)/2\alpha_{ij}^2$.

**General case:** With this, and by triangle inequalities, $Y_j \leq X_j + Z_j$ and $X_i \leq Y_i + Z_i$ and $Y_i + Y_j \geq \alpha$, we get

$$
\begin{aligned}
\Pr[h_{f,S}(x_i) = j] &= \Pr[X_j \leq X_i] \\
&\leq \Pr[Y_j \leq Y_i + Z_i + Z_j] \\
&\leq \Pr[Y_j \leq \tfrac{3}{4}\alpha \ \lor \ Y_i + Z_i + Z_j \geq \tfrac{3}{4}\alpha] \\
&\leq \Pr[Y_i \geq \tfrac{\alpha}{4} \ \lor \ Y_i \geq \tfrac{\alpha}{4} \ \lor \ Z_i + Z_j \geq \tfrac{\alpha}{2}] \\
&\leq \Pr[Y_i \geq \tfrac{\alpha}{4}] \ + \ \Pr[Z_i + Z_j \geq \tfrac{\alpha}{2}] \\
&\leq \Pr[Y_i^2 \geq \tfrac{\alpha^2}{16}] \ + \ \Pr[Z_i^2 + Z_j^2 \geq \tfrac{\alpha^2}{8}] .
\end{aligned}
$$

Now by Markov's inequality,

$$
\Pr[Z_i^2 + Z_j^2 \geq \tfrac{\alpha^2}{8}] \ \leq \ \mathbb{E}[Z_i^2 + Z_j^2]/(\tfrac{\alpha^2}{8}) \ = \ \frac{8\sigma_i^2 + 8\sigma_j^2}{n_c \alpha_{ij}^2} ,
$$

and similarly $\Pr[Y_i^2 \geq \frac{\alpha^2}{16}] \leq 16\sigma_i^2/\alpha_{ij}^2$. Plugging this into (36), by symmetrization, leads to the desired inequality

$$
\mathbb{E}[\mathrm{Err}] \ \leq \ \frac{1}{k} \sum_{i \neq j} \frac{16\sigma_i^2}{\alpha_{ij}^2} + \frac{8\sigma_i^2 + 8\sigma_j^2}{n_c \alpha_{ij}^2} \ = \ 16(k-1)(1 + \tfrac{1}{n_c}) \operatorname{Avg}_{i \neq j}[V_f(P_i, P_j)] .
$$

**Spherically symmetric case:** In this case

$$
\begin{aligned}
\Pr[h(x_i) \neq j] &= \Pr[||u_i - \hat\mu_j|| \leq ||u_i - \hat\mu_i||] \\
&= \Pr[u_i \in H] \ = \ \Pr[d(u_i, H) = 0], \quad \text{where}
\end{aligned}
$$

$$
\begin{aligned}
H &:= \{x \in \mathbb{R}^p : ||x - \hat\mu_j|| \leq ||x - \hat\mu_i||\} \\
&\equiv \{x \in \mathbb{R}^p : \langle \hat\mu_i - \hat\mu_j, \ 2x - \hat\mu_i - \hat\mu_j \rangle \geq 0\} \\
&\equiv \{x \in \mathbb{R}^p : d(x, H) = 0\}
\end{aligned}
$$

is the half-space of all $u_i$ that are wrongly classified, and $d(x, H)$ is the distance of $x$ from separating hyperplane $\partial H$ if $x \notin H$ and 0 if $x \in H$. The distance $D$ of the closest point of $H$ to $\mu_i$ is

$$
D \ = \ d(\mu_i, H) \ = \ \max\left\{0, \left\langle \frac{\hat\mu_i - \hat\mu_j}{||\hat\mu_i - \hat\mu_j||}, \ \mu_i - \frac{\hat\mu_i + \hat\mu_j}{2} \right\rangle\right\}
$$

For fixed displacement magnitudes $Z_i$ and $Z_j$, this distance is smallest if both displacements are colinear in direction of $\mu_i - \mu_j$, in which case

$$
D \ \geq \ D_{min} \ := \ \frac{\alpha}{2} - \frac{Z_i + Z_j}{2}
$$

A formal proof of this claim is as follows: W.l.g. we can choose (a coordinate system such that) $\mu_i = 0$:

$$
\begin{aligned}
2D &= -\frac{\langle \hat\mu_i - \hat\mu_j, \ \hat\mu_i + \hat\mu_j \rangle}{||\hat\mu_i - \hat\mu_j||} = \frac{||\hat\mu_j||^2 - ||\hat\mu_i||^2}{||\hat\mu_i - \hat\mu_j||} = (||\hat\mu_j|| - ||\hat\mu_i||)\frac{||\hat\mu_j|| + ||\hat\mu_i||}{||\hat\mu_i - \hat\mu_j||} \\
&\geq ||\hat\mu_j|| - ||\hat\mu_i|| \ \geq \ ||\mu_j|| - ||\hat\mu_j - \mu_j|| - ||\hat\mu_i|| \ = \ \alpha - Z_j - Z_i \ = \ 2D_{min} ,
\end{aligned}
$$

where both inequalities are simple applications of the triangle inequality.

Now, for fixed $\hat\mu_i$ and $\hat\mu_j$ i.e. fixed $D$ and by rotational invariance, the probability that $u_i \in H$ is the same as the probability that (e.g.) the first coordinate of $u_i - \mu_i$ is larger than $D$. With $Y_i^1 := u_i^1 - \mu_i^1$ and $\gamma \in (0, 1)$, this implies

$$
\begin{aligned}
\Pr[u_i \in H] &= \Pr[Y_i^1 \geq D] \\
&= \Pr[Y_i^1 \geq D \ \land \ Z_i + Z_j \leq \gamma\alpha] \ + \ \Pr[Y_i^1 \geq D \ \land \ Z_i + Z_j > \gamma\alpha] \\
&\leq \Pr[Y_i^1 \geq 0.5(1-\gamma)\alpha] \ + \ \Pr[Z_i + Z_j > \gamma\alpha] \\
&\leq \Pr[(Y_i^1)^2 \geq 0.25(1-\gamma)^2\alpha^2] \ + \ \Pr[Z_i^2 + Z_j^2 > 0.5\gamma^2\alpha^2] \qquad (37) \\
&\leq \frac{\mathbb{E}[(Y_i^1)^2]}{0.25(1-\gamma)^2\alpha^2} + \frac{\mathbb{E}[Z_i^2 + Z_j^2]}{\gamma^2\alpha^2} \\
&= \frac{\sigma_i^2/p}{0.25(1-\gamma)^2\alpha^2} + \frac{(\sigma_i^2 + \sigma_j^2)/n_c}{\gamma^2\alpha^2} . \qquad (38)
\end{aligned}
$$

Plugging this into (36) with $\gamma = 0.5$, by symmetrization, leads to

$$\begin{aligned}
\mathbb{E}[\text{Err}] &\leq \frac{1}{k} \sum_{i \neq j} \frac{8\sigma_i^2 + 8\sigma_j^2}{\alpha_{ij}^2} \left[ \frac{1}{p} + \frac{1}{n_c} \right] \\
&= 16(k-1) \left[ \frac{1}{p} + \frac{1}{n_c} \right] \text{Avg}_{i \neq j} \left[ V_f(P_i, P_j) \right]
\end{aligned}$$

$\square$

