# OpenReview forum: "On the Role of Neural Collapse in Transfer Learning"
_ICLR.cc/2022/Conference — ICLR 2022 Poster_

### Official Review · Reviewer_QSqm · 2021-10-30

**Correctness:** 4
**Technical Novelty And Significance:** 2
**Empirical Novelty And Significance:** 3
**Recommendation:** 6
**Confidence:** 4

**Main Review:**

Strength:
1) The experiments are done neatly, I appreciate the amount of hard works of large scale computing to generate fig 1., and their experimental results supports their argument very well.
2) Study Neural collapse on unseen classes for transfer learning is new
3) The writing and flow of the paper is very clear, especially the statement of their transfer learning's math setting are pretty neat.
4) Their own version of neural collapse definition via class-distance normalized variance (CDNV) is good, although it is kind of a natural rew derivative quantity of
 (a) the signal to noise ratio in the original NC papers (for eg. https://arxiv.org/pdf/2106.02073.pdf).
 (b) the notion of separation in NC by https://arxiv.org/pdf/2012.10424.pdf

Weaknesses:
1) The theory analysis section(section 4) tries to supports the experiments, as a faithful decoration to an experimental paper. It did a decent job, but the analysis is not too innovative or insightful.

  Feedback to improve:
  (a) It would be good if the paper can provide a more in-depth analysis of why NC works for some unseen classes in transfer learning in terms of the distance between unseen classes and the classes that are learned in original train dataset for foundation models in semantic sense.
  (b) It would be good if the paper can provide a mathematical analysis that touch the inside of deep neural network rather than treat it as embedding and only study the last linear layer
(c) study beyond the standard techniques to derive generalization bounds to new class-conditional distributions that tightly capture deep neural network's representation power.

2) On experimental side, all the measurement are done with x axis being the course of training iterations. Another natural x-axis for measurement of interest include:
(1) number of transfer dataset sample size per class
(2) number of train dataset sample size per class

(2) would probably require another works but I think (1) is related to the paper's claim on few-shot learning.




**Summary Of The Paper:**

Summary:
The paper study neural collapse for classification in transfer learning setting, where the large and overparameterized foundation models trained over many classes could provide feature maps that are transferable to new, unseen classes with few-shot learning .

The paper demonstrate both theoretically and empirically that neural collapse generalizes to new samples from the training classes, and to new classes.

Contributions:
They demonstrate that neural collapse generalizes to new data points and new classes (e.g., the target classes), when the model is trained on a large set of classes with many samples for each class. In addition, they show that in the presence of neural collapse, training a linear classifier on top of the learned penultimate layer can be done using only few samples.

On experimental side, the experiments on neural collapse in transfer learning is new to my knowledge. And the experiments are at large scale, with clear organizations.

On theory aspect, the paper's contribution is mostly to accompany and support the experiments.


**Summary Of The Review:**

The paper did solid experiments on Neural collapse on unseen classes for transfer learning. The angle is new and the experiments are done neatly with a good amount of hard works of large scale computing. And their experimental results supports their main claim very well. The theory part is more standard and I have not found surprising theoretical insights from it. Overall it is a good paper worth recommending to ICLR as a solid incremental work in the field of Neural collapse.

---

> ### Author Response · Authors · 2021-11-23
> **Response to reviewer QSqm**
>
> > Reviewer: The proposed CDNV is good, but not novel.
>
> Answer: We agree that the proposed CDNV is not novel; it was introduced as an alternative to CCNV to facilitate the analysis. Our contribution lies in connecting two disjoint lines of work (neural collapse and few-shot learning with foundation models) and to explain the recent success of foundation models in few-shot transfer learning (see the paragraph “Contributions” in Section 1).
>
> > Reviewer: The theoretical analysis is decent but not too innovative or insightful.
>
> Answer: We agree the analysis is not innovative as we do not intend to design or introduce new proof techniques. However, we politely disagree with the claim that the analysis is not insightful. In fact, neural collapse has been established in the literature only for the training data. In our analysis we show that this generalizes to test data from the same classes (Proposition 1), test data from new classes (Proposition 2), and that it is tightly related to few-shot learnability (Section 4.3). When summarizing the analysis in Sections 4.1-4.3 we conclude that training a foundation model on a large set of classes reduces the sample complexity of learning new classes. This is an intriguing property that has been observed empirically, with no theoretical understanding behind it. In this paper, we provide a theoretical understanding for the mechanisms that allow that.
>
> > Reviewer: A more in-depth analysis dependent on the distance/relationship between the target and source classes.
>
> Answer: We agree that to ensure that NC generalizes to new classes, we certainly need to assume that the source and target classes are related in one way or another. If the classes are extremely unrelated we should not expect the foundation model to be useful for the target classes. For this purpose, as we describe in Section 4.2, we assume that the classes are sampled from a given distribution $\mathcal{D}_C$ (subject to being different from one another). This allows us to guarantee that if NC occurs over the source classes it would be translated to target classes. This i.i.d.(-like) assumption allows us to treat the samples/tasks/classes as coming from the same ‘pool’ of samples/tasks/classes. This is a standard assumption in learning theory [1,2].
>
> We agree that analyzing the performance in terms of class similarity is an interesting question, but it is not entirely clear how the similarity should be defined (it would be easy to define it in terms of the learned feature map $f$, but then we would start from some assumptions if $f$ is good or bad for the new classes, taking away the interesting questions about the generalization of $f$ to unseen classes).
>
> It is unclear to us what you mean by ‘semantic sense’. We would be grateful if you could please clarify that point.
>
> [1] V. Vapnik. “Statistical Learning Theory.” 1989.
> [2] S. Shalev-Schwartz and S. Ben-David. “Understanding Machine Learning: From Theory to Algorithms.” 2014.
>
> > Reviewer: “... provide a mathematical analysis that touch the inside of deep neural network rather than treat it as embedding and only study the last linear layer.”
>
> Answer: Understanding the training dynamics and generalization in deep networks is a highly complicated task and is a long-standing open problem in the field. In this work we do not intend to fully resolve this problem, and describe generalization and dynamics in deep learning. Rather, we develop a modular argument that builds upon the presence of the empirical phenomenon of neural collapse to provide a new angle for understanding the puzzling problem of generalization in deep learning, especially in the few-shot setting. As we show, the emergence of neural collapse in the top embedding layer is tightly related to the ability to generalize to new classes. As mentioned in Section 3, studying the conditions and dynamics in which this property emerges is done in other works.
>
> > Reviewer: Examining the performance as a function of the (1) number of target samples and (2) number of training samples.
>
> Answer: In the revised version we provide new experiments for (1) (see Table 1 and Figures 4-5). While (2) is also an interesting question in general, in our context the effect of the number of samples per class in the training dataset would show up through the strength of the neural collapse observable in the test data. We touch this question analytically in Proposition 1 and the related analysis of the $\epsilon$-terms in Proposition 3 in Appendix E, but we believe that the empirical analysis of this question is out of the scope of the paper.

---

### Official Review · Reviewer_dW2d · 2021-11-02

**Correctness:** 3
**Technical Novelty And Significance:** 1
**Empirical Novelty And Significance:** 3
**Recommendation:** 6
**Confidence:** 3

**Details Of Ethics Concerns:**

No ethics concerns

**Main Review:**

Strengths:
+ This paper is well-written, and it is enjoyable to read.
+ Neural collapse is only investigated existing samples and classes before. It is significant that this paper shows that neural collapse generalizes to new samples from the training classes and new classes both theoretically and empirically.
+ Comparisons to prior work validate the views of this paper in some respects.

Some concerns:
+ It is too idealistic to only consider a two-class classification problem in section 4.3.  The real classification problems are often multiclass classification problems. Can the conclusion in section 4.3 be extended to multiclass classification problems?
+ Neural collapse is an emergent phenomenon in deep learning. In the few-shot settiing, is it related to the architecture of the network? More architectures (e.g. VGG or Inception) will be appreciated.
+ It is known that learning rate strategies in SGD need to be carefully tuned. If other optimizers (e.g. adam) are used, are the experimental conclusions consistent?
+ It is shown that neural collapse occurs when training neural networks under MSE-loss minimization with different settings of regularization. Are the experimental conclusions consistent under MSE-loss minimization?  Also, it is very interesting to find the role of neural collapse for regression problems (MSE-loss minimization is commonly used in regression) in the few-shot setting.
+ Are the conclusions of this paper consistent in other areas, for example, NLP?

**Summary Of The Paper:**

This paper studies the ability of foundation models for classification to learn representations that are transferable to new, unseen classes in the perspective of neural collapse. Specifically, this paper demonstrates both theoretically and empirically that neural collapse generalizes to new samples from the training classes and new classes. As a result, representation learned by foundation models can be easily classified in the few-shot setting.

**Summary Of The Review:**

Neural collapse is only investigated existing samples and classes before. It is significant that this paper shows that neural collapse generalizes to new samples from the training classes and new classes both theoretically and empirically. However, there are some minor concerns and I think this paper is marginally above the acceptance threshold.

---

> ### Author Response · Authors · 2021-11-23
> **Response to reviewer dW2d**
>
> > Reviewer: Can the conclusion in section 4.3 be extended to multiclass classification problems?
>
> Answer: In the original submission we chose to demonstrate the analysis in Section 4.3 in a simple way, for binary classification only. The results can be extended to multi-class classification using the union bound in a straightforward way; in the revised paper we give a detailed derivation of all the results in this section in Proposition 5 in Appendix G. The extension to multiclass classification is a straightforward application of union bound.
>
> > Reviewer: Is neural collapse architecture-dependent?
>
> Answer: To provide a more conclusive report, we already conducted an experiment with a convolutional network (see, for example, Fig. 1 second row) in the submitted version of the paper. Furthermore, in the paper introducing neural collapse, Papyan et al. (2020) already provided an extensive set of experiments showing that the CCNV (on the training data) significantly decreases during training on MNIST, FashionMNIST, CIFAR10, CIFAR100, ImageNet, STL10 using VGG, ResNet and DenseNet. Since our CDNV is just a slightly modified version of their CCNV (to facilitate theoretical analysis but keeping the main qualitative properties of CCNV), we expect to see similar behaviour with our metric in other related settings (we also ran additional experiments with other wide resnets and resnets, where we could observe neural collapse with CDNV).
>
> > Reviewer: Are the conclusions consistent when changing the optimization method?
>
> Answer: As mentioned in paragraph “Architectures and hyperparameters” we validated the consistency of our results across different choices of learning rates, see Figures 2-3 and Appendix B.1. During the rebuttal, we added experiments with learning rate scheduling, see Figure 5 and Appendix B.1.
>
> > Reviewer: Are the experimental conclusions consistent under MSE-loss? What is the role of neural collapse in regression?
>
> Answer: Our theoretical results only depend on the assumption that neural collapse happens during training, hence in this sense they are independent of the actual loss used during the training. Even though neural collapse also emerges when training with the networks with MSE (as demonstrated extensively by Han et al., 2021), training with the cross-entropy (CE) loss typically achieves better performance in classification problems. Therefore, we chose to experiment with CE. However, we also expect that similar conclusions about the applicability of foundation models would hold when they are trained with MSE as well (depending on what level of neural collapse is achieved in training).
>
> Regarding regression, neural collapse defines a learning regime in which the embeddings of samples from the same class tend to have a small variance relative to the distances between their class means. Since this notion is tightly related with the presence of classes in the data, it is not clear to us how this concept can be extended to regression, and we are unaware of any similar work in the literature. Nevertheless, it is an interesting question if neural networks can learn embeddings which similarly simplify the solution of regression problems as neural collapse does for classification.
>
> > Reviewer: Are the conclusions of this paper consistent in other areas, for example, NLP?
>
> Answer: Papyan et al. (2020) observed the presence of neural collapse in several well-established visual classification problems. Since we study the connection between neural collapse and transfer learning, we experimented with similar datasets as they did. To our knowledge, the concept of neural collapse has not been considered in the context of NLP, and hence at the moment it is not clear that the success of large language models in few-shot learning tasks (such as GPT-3) has any connection with some variant of neural collapse in the embeddings of language models. Studying the emergence of neural collapse in NLP models is an interesting problem, but it is out of the scope of this paper.

---

### Official Review · Reviewer_e2QG · 2021-11-02

**Correctness:** 3
**Technical Novelty And Significance:** 3
**Empirical Novelty And Significance:** 3
**Recommendation:** 8
**Confidence:** 3

**Main Review:**

#### Strengths
 - The paper provides a possible explanation as to why representations learned over many classes by a lone classifier can serve as a competitive solution to few-shot learning problems, by exploring the idea of neural collapse.
 - Overall the explanation(hypothesis) provided for the success of foundation models in few shot settings due to NC, is well-motivated by the authors and the general flow of the paper makes sense.
 - The theoretical justifications about NC occurring in unseen classes, and therefore translating to better accuracy in few-shot tasks seem like a novel contribution that gives an insight into the reasons a single classifier works well in a few shot problems when compared to algorithms devised specifically for such tasks.
- It is good to note that experiments have averaged reported performance over various random initialization and the last few epochs of training to ensure that results are more robust.

#### Weaknesses
- The performance of the solution proposed is not the best and is especially far from it in the case of MiniImageNet. While I understand that the paper is not focused on beating the SOTA (Tian et al.) but rather develops on the insight that they gather from neural collapse, I don't agree with the reasoning provided for it. The authors attribute other methods' higher performance to sophisticated initialization, scheduling schemes, or distillation procedures, etc. I don't fully agree since the performance gap is not very small to be attributed to such reasons. For instance, the performance of (Tian et al.)'s simple Baseline model(not even the distilled model) performs much better on 5-shot 5-way MiniImagenet (79.64 ± 0.44).
- The experiment setting chosen can be thought to be similar to 5-way 5-shot problems. An important experiment to demonstrate their solution would be a 5-way(or n-way) 1-shot setting. Since the paper claims the competitiveness of foundation models on such tasks over some specialized algorithms, it is important that results on 1-shot tasks are also reported on as similar works have also been done. For example, (Tian et al.)'s simple baseline model (which works on the principle of pretraining like foundation models) has reported SOTA scores on 5-way 1-shot MiniImagenet (73.9 ± 0.8).

[2] Tian, Yonglong, Yue Wang, Dilip Krishnan, Joshua B. Tenenbaum, and Phillip Isola. "Rethinking few-shot image classification: a good embedding is all you need?." In Computer Vision–ECCV 2020: 16th European Conference, Glasgow, UK, August 23–28, 2020, Proceedings, Part XIV 16, pp. 266-282. Springer International Publishing, 2020.

**Summary Of The Paper:**

Large pretrained models that can be effectively adapted to various tasks via transfer learning have been recently characterized as foundation models. (Papyan et al.) recently introduced the idea of neural collapse(NC) in deep networks. It identifies the training dynamics of deep neural networks for classification tasks, where the penultimate layer features associated with training samples from the same class concentrate around their class' feature means.
        The authors recognize the recent results that show the competitive performance of a single classifier trained over many classes on few-shot learning problems. They try to provide an explanation for such behavior based on NC. They theoretically show that if NC happens on a training set, then it generalizes to unseen samples from the same task under natural assumptions.
        Second, they show that NC is expected to happen in new classes when classes for pretraining(source tasks) and finetuning(target tasks) come from the same class distributions. This would imply that representations learned during training are adaptable to other tasks. To demonstrate the above results, authors work with class-distance normalized variance(CDNV) to identify NC in contrast to (Papyan et al.) that use Class-Covariance Normalized Variance (CCNV). The intuitive explanation for using CDNV has been described by the fact that when the variance of features of a certain class decreases in comparison to distance from another class, one expects a higher accuracy for classifying features belonging to different classes.
        Finally, the authors study the relation between CDNV and the classification performance and show that low CDNV corresponds to low error probability, thereby explaining the success of foundation models in few-shot settings when NC occurs. The authors also validate their theoretical results empirically, by demonstrating that NC is strongly correlated with accuracy in few-shot settings.

[1] Vardan Papyan, X. Y. Han, and David L. Donoho. Prevalence of neural collapse during the terminal phase of deep learning training. Proceedings of the National Academy of Sciences, 117(40): 24652–24663, 2020.

**Summary Of The Review:**

 Even though there are some concerns about the experiments in the paper, I believe the paper has more merits than flaws, in proposing a possible explanation for the superior performance of foundation models in few-shot settings, and validating it theoretically and empirically.

---

> ### Author Response · Authors · 2021-11-23
> **Response to reviewer e2QG**
>
> > Reviewer: Difference from SOTA
>
> Answer: We agree with the reviewer that the difference is probably not because of sophisticated tuning, etc., methods. Our current hypothesis is that the main difference is the classification method used on top of the feature layer. We use linear regression, while the best method of Tian et al. (2020) uses logistic regression. In their Table 5, they also present results with nearest neighbor classification, and our results are better than those (this is explained in footnote 1 on page 9 in the revised paper). During the revision we applied some simple tuning methods (learning rate scheduling, model selection), which somewhat improved the performance of our method (we have SOTA results on FC-100 1-shot 5-class, even without these tricks), see Table 1 in the revised paper. Due to the lack of time, we were not able to verify this hypothesis experimentally -- we experimented with different architectures, keeping the linear regression part unchanged, but did not manage to reproduce the results of, e.g., Tian et al. (2020).
>
> Even though our results are not (always) SOTA, we believe that the conclusions we draw are meaningful.
>
> > Reviewer: 1-shot experiments
>
> Answer: In the revision we added experiments with varying the number of samples in the few-shot tasks, considering 1, 5, 10, 20-shot performance (Figures 4-5). A detailed comparison with the literature on 1- (and 5-) shot problems is given in Table 1. One can see that our method is quite competitive on 1-shot problems, even reaching SOTA on FC-100. We would like to mention that contrary to the statement of the reviewer, the simple method of Tian et al does not achieve SOTA on 1-shot 5-way MiniImagenet: they achieve 62.02 ± 0.63 and not  73.9 ± 0.8, which is in fact the performance of their better method on the easier 1-shot 5-way CIFAR-FS (see Tables 1 and 2 in their paper, or Table 1 in ours)
>
> As a side note, we mention that to get a better comparison with Tian et al. (2020), we reproduced the accuracy rates to be averaged between epochs 90 and 100 (in their paper they run until epoch 90) -- the accuracy rates we obtain this way are very similar to the old ones.

---

> > ### Comment · Reviewer_e2QG · 2021-11-29
> > **Thanks for the additional results**
> >
> > I have gone through the newer results which I had requested. Overall the empirical evaluation looks more robust now with good performance of the authors' approach in 1 shot settings as well. While there isn't much novelty in the theoretical tools used for various proofs, I appreciate the idea of providing an explanation for the success of foundation models in few-shot settings and justifying it as well. Due to this,  I change my recommendation to
> > 8: accept, good paper

---

### Official Review · Reviewer_XpXZ · 2021-11-03

**Correctness:** 3
**Technical Novelty And Significance:** 3
**Empirical Novelty And Significance:** 2
**Recommendation:** 6
**Confidence:** 3

**Main Review:**

Strength:
1. This paper provides some theories on the neural collapse in the transfer learning setting

Weakness:
The theories are not sufficient, see details below:
1. On page 4, before Section 4, there is a claim: "Our version of neural collapse (at training) asserts that lim_{t\to\infty} Avg V_f(S_i, S_j) =0". Why is this true? This paper doesn't give any justification from either the theoretical or numerical perspective.
2.  Proposition 1 basically claims that the empirical and population version of V_f are close to each other, which is a variant of generalization bound. The establishment of this generalization bound is the key part in this kind of statement, however, this paper neglects this part by directly assuming there exists a generalization bound for the first and second moments of f, which makes the theoretical contributions not sufficient.
3. Proposition 2 is more interesting, but the current results are established with respect to the expectation, and the RHS does not depend on the sample size at all. A more refined analysis that shows explicit dependence on the sample size (and the number of source tasks) would help.
4. It would be better if we have a formalized theorem for the classification error in Section 4.3.



**Summary Of The Paper:**

This paper analyzes neural collapse in the transfer learning setting using foundation models.

**Summary Of The Review:**

This paper claims to provide theoretical insights on the neural collapse in the transfer learning setting, but the theoretical results are not sufficient.

---

> ### Author Response · Authors · 2021-11-23
> **Response to reviewer XpXZ**
>
> > Reviewer: Why do you assert that $\lim_{t\to\infty} Avg_{i\neq j} V_f(S_i, S_j) =0$?
>
> Answer: First, note that our theoretical results (Propositions 1-2) do not require that this quantity tends to zero, rather, we show that if this quantity is small, then, the classification performance is good (Section 4.3 and Proposition 5).
>
> On the other hand, our CDNV is essentially the same as the CCNV used by Papyan et al. (2020) (we updated Section 3 to address that), with the benefit that it simplifies our theoretical analysis. In their paper, Papyan et al. provided a wide range of experiments showing that the CCNV significantly decreases during training on MNIST, FashionMNIST, CIFAR10, CIFAR100, ImageNet, STL10 using VGG, ResNet and DenseNet. As discussed by Papyan et al. (2020), this is especially pronounced when the optimization converges to a solution that has zero train classification error. We empirically observe how the CDNV decreases in several experiments, reported, for example in Figures 1 and 2 (the decrease is more pronounced for smaller learning rates as it allows to interpolate the whole source training dataset, see, e.g., Figure 2).
>
> On the theoretical side, most contributions (Papyan et al., 2020; Zhu et al., 2021; Mixon et al., 2021; Poggio & Liao, 2020a;b) identify settings in which neural collapse holds for training, or the feature map $f$ induced by the global minima of the training loss satisfies $Avg_{i\neq j} V_f(\tilde{S}_i, \tilde{S}_j)=0$ (for large enough sample sizes).
>
> We updated the end of Section 3 (after the definition of neural collapse) with the explanations above.
>
> > Reviewer: The paper assumes generalization bounds for the first and second moments of $f$ instead of explicitly proving those.
>
> Answer: We added two propositions (in Appendix E)  to the revised version of the paper to address this. Proposition 3 introduces generalization bounds for the first and second moments of $f$. In addition, we added Proposition 4 which shows that the Rademacher complexity in Proposition 2 scales as $O(\sqrt{l})$, and therefore, the generalization gap scales as $O(1/\sqrt{l})$. We also streamlined the statements and proofs of the existing propositions.
>
> > Reviewer: Proposition 2 is established with respect to the expectation and the RHS does not depend on the sample size.
>
> Answer: In the revised version we updated the statement of Proposition 2 to be a high probability bound, instead of a bound that holds in expectation (in the previous version we proved Proposition 2 based on Theorem 2 of Maurer and Pontil, 2019, while in the current version we employ its high probability version -- their Corollary 3).
>
> Our overall analysis is split into two parts. In the first part (Proposition 1) we upper bound the CDNV between a pair of source classes $\tilde{P}_i,\tilde{P}_j$ in terms of the CDNV between the source datasets $\tilde{S}_i,\tilde{S}_j$. This bound depends on the datasets $\tilde{S}_i,\tilde{S}_j$ and their sizes $m_i,m_j$ through $\epsilon^c_1, \epsilon^c_2$ ($c=i,j$). In the second part (Proposition 2) we estimate the expected CDNV between two new unseen classes in terms of the averaged value of the CDNV between the source classes $\tilde{P}_c$. We can combine the two propositions trivially using a simple union bound, and upper bound the CDNV of the class distributions in Proposition 2 by the empirical CDNV by Proposition 1. This gives an upper bound on the expected CDNV that depends on (a) the CDNV between the source datasets $\tilde{S}_c$, (b) the size $m_c$ of $\tilde{S}_c$ and (c) the number of source classes $l$.
>
> > Reviewer: Formalize the analysis in Section 4.3.
>
> Answer: We added Proposition 5 (Appendix G) that summarizes the analysis in Section 4.3. For completeness, we extended the analysis to $k$ target classes instead of 2 (by using a union bound). In the current version, we refer to this proposition in Section 4.3.
>
> > Reviewer: Insufficiency of the theoretical results.
>
> Answer: To complement our existing results, during the rebuttal we added a number of precise statements about missing details and streamlined the existing derivations. The new parts include (a) a formal analysis for Section 4.3 for the multiclass case (Proposition 5 in Appendix G); (b) upper bounds for the $\epsilon^c_i$ terms for ReLU neural networks (Proposition 3 in Appendix E) and for the Rademacher complexity term in Proposition 2 (Proposition 4 in Appendix E). We also updated the bound in Proposition 2 to hold with high probability instead of expectation. Combining all these results show that with high probability, neural collapse in training implies a small error in classification with new tasks. We now mention this in Section 4.3.

---

> > ### Comment · Reviewer_XpXZ · 2021-11-27
> > **thanks for the explanation**
> >
> > I appreciate the authors' explanation, but I am still confused by a few points.
> >
> > 1. All I want is some justifications when you claim something in the paper. My point is that if you say in the paper that your version of neural collapse is $\lim_{t\to\infty} Avg_{i\neq j}V_f(S_i, S_j)=0$, you should justify this by either some theoretical derivations or numerical experiments. Such justification seems lacking in the current revision. Additionally, in the revision, the authors added more sentences "most contributions (Papyan et al., 2020; Zhu et al., 2021; Mixon et al., 2021; Poggio & Liao, 2020a;b) identify settings in which neural collapse holds for training, or the feature map  induced by the global minima of the training loss satisfies $\lim_{t\to\infty} Avg_{i\neq j}V_f(S_i, S_j)=0$ (for large enough sample sizes)." Again, such a claim is not clear to me, and more theoretical derivations should be added to the supplement to support this claim.
> >
> > 2. Regarding the updated version of Proposition 2, I assume the authors have already spent enough carefulness and effort to revise the original results. However, in this current statement of Proposition 2 in the revision, the LHS is still in expectation?
> >
> > 3. My another main concern is the novelty of the paper. Neural collapse is an interesting phenomenon observed in the training samples. The main contribution of this paper is the claim that such a phenomenon also holds in the test sample (under two settings: 1. the samples are iid; 2. the distributions are iid, assuming the distributions are drawn from a distribution of distributions). However, the justification of such a claim seems not novel in my current mind. In the original version, the paper simply **assumes** the gap between the training and test samples are small. Under this assumption, the claim holds in a straightforward way. In the revision, this paper bounds the gap between the training and test samples by standard Rademacher argument. The techniques here are not novel either.
> >
> > Given the above comments, I will maintain my score for now. I am willing to increase the score though if the authors can add more convincing justifications, especially on my third point.

---

> > > ### Author Response · Authors · 2021-11-28
> > > **Clarifications (part 2)**
> > >
> > > **Re Comment 2**
> > >
> > > Our final goal (combining the results from Section 4.1-Section 4.3) is to give a bound on the expected error of the learned classifier on new samples from the few-shot problems: as the classifier (and the feature map $f$) is a function of the training data, this expectation is in fact a conditional expectation given the training data (this shows up only implicitly in our notation, since we show over which variables we take expectation over, not the conditioning variables). The dependence on the training data is typically handled in two ways in learning theory: either by taking expectation over the training data, or proving high-probability bounds where the bound on the expected error (given the training data) holds with high probability over the (random) choice of the training data. Our argument is of the latter type.
> > >
> > >
> > > Formally, our argument (which is also the final argument in Section 4.3 in the revised paper) can be detailed as follows:
> > >
> > >
> > > The expected few-shot error of a fixed feature map $f$ is defined as
> > >
> > > $$err(f) = E_{P_1,...,P_k} E_{\forall c \\in [k]: S_c \\sim P^{n_c}\_c} E_{(x,y) \\sim P} \\Pr[h_{S,f}(x) \neq y]$$
> > >
> > > where $P_1,...,P_k \sim \mathcal{D}\_C(P\_1,...,P\_k | \\forall i\\neq j \\in [k]: P_i\\neq P_j)$ and $h\_{S,f}$ is a classifier computed based on $f$ and the random training data $S_1,...,S_k$.
> > > In this definition expectation is taken over the random selection of the few-shot problem, as well as the random training samples in the selected few-shot problem.
> > > In Proposition 5, we upper bound $E_{\forall c \\in [k]: S_c\\sim P^{n_c}\_c} E_{(x,y)\\sim P} \\Pr[h_{S,f}(x) \neq y]$
> > > for a fixed set of target classes $P_1,...,P_k$ and feature map $f$.
> > >
> > > In order to upper bound the expected few-shot error $err(f)$ on randomly selected target classes $P_1,...,P_k$, we can simply take expectation over $P_1,...,P_k$ on both sides of the inequality in Proposition 5.
> > > By doing so, we can bound the expected few-shot error $err(f)$ (this is the quantity we care about).
> > > The expectation of the upper bound in Proposition 5 becomes proportional to the expected value of the CDNV between pairs $P_i\neq P_j$. This is exactly the term $E\_{P_{i}\\neq P_{j}}[V_f(P_i,P_j)]$ on the left-hand side in Proposition 2.
> > >
> > > Then, by applying Proposition 2, we can bound this term with the averaged CDNV between the source classes, $Avg_{i\\neq j}[V_f(\tilde{P}_i,\tilde{P}_j)]$ (plus additional terms that arise from Proposition 2), which holds with high probability over the selection of the training classes for the foundation model.
> > > By Proposition 1 we can bound this quantity in terms of $Avg\_{i\\neq j}[V_f(\tilde{S}\_i,\tilde{S}\_j)]$
> > > (by Proposition 3, the conditions with certain choice of the $\epsilon^c_i$ functions hold with high probability over the choice of the training samples).
> > > By combining these bounds, we can bound the expected few-shot error $err(f)$ of $f$ (given the training data for the foundation model which is used to compute $f$) in terms of
> > > $Avg\_{i\\neq j} [V\_f(\tilde{S}\_i,\tilde{S}\_j)]$, the number of source classes $l$ and the sizes $m_c$ of $\tilde{S}\_c$, as well as the number of target classes $k$ and the number of samples $n_c$ in the few-shot task. The conclusion is that if $Avg\_{i\\neq j} [V\_f(\tilde{S}\_i,\tilde{S}\_j)]$ gets small when training $f$ as part of training a classifier on the source classes (i.e., neural collapse emerges during train time) and $l$ and $m_c$ are large enough, we expect $err(f)$ to be small (with high probability over the selection of the training data and source classes).
> > > Note that this argument does not require that $Avg\_{i\\neq j} [V\_f(\tilde{S}\_i,\tilde{S}\_j)]$ would tend to 0. We will emphasize these points in the final version of the paper, and we can summarize this analysis in a formal statement (in the appendix).

---

> > > ### Author Response · Authors · 2021-11-28
> > > **Clarifications (part 1)**
> > >
> > > Thanks for continuing the discussion. Below we provide explanations to your comments (first on 1 and 3, and in the second part, on 2), which we will incorporate in the paper. Please do not hesitate to get back to us should you need more explanations.
> > >
> > > **Re Comment 1:**
> > >
> > > The CDNV we introduce is essentially the same as the CCNV considered in previous work (see, e.g., Figs. 8-11), hence the previous findings in the literature apply to our definition of neural collapse. Previous work (e.g., Fig. 6 of Papyan et al., 2020), and also ours (e.g., Figs. 1-5), find experimentally  that CCNV/CDNV decreases throughout training, and hence Papyan et al. (2020) defines neural collapse as CCNV converging to zero, and we followed this definition in our paper.
> > > On the other hand, we realize that the CDNV/CCNV values we have are not extremely small (neither are they for ImageNet and CIFAR100 for Papyan et al.), so in this sense we agree with the reviewer that the experiments do not fully support a claim that CDNV/CCNV converge to zero. Note, however, that in our experiments we usually do not train until 0 training error (to avoid overfitting to the selected small number of source classes in training the the feature map $f$), and when we do, e.g., for smaller learning rates in Fig. 2, the CDNV also gets smaller -- in fact this interpolation regime is where we expect CDNV/CCNV to vanish. Nevertheless, we do not claim the convergence of CDNV to zero -- we only use it to define "neural collapse", following the existing definition in the literature. All our results hold in general, and are already meaningful for reasonably small CDNV values, which indeed show up in the experiments. We are happy to replace our (only) mention that CDNV converges to zero at the end of Section 3 with saying that it becomes small.
> > >
> > > On the theoretical side, the existing results arguing that neural collapse shows up in training under some restricted conditions (e.g., linear regression), are also applicable for our neural collapse definition (with CDNV in place of CCNV). For example, Theorem 3.1 of Zhu et al. (2021) shows that any unconstrained feature map minimizing a cross-entropy loss with some regularization maps each (training) sample from a class to the same feature vector, in which case CDNV and CCNV are in fact 0 (unconstrained feature maps are approximations to large overparametrized neural nets).
> > >
> > >
> > > **Re Comment 3:**
> > >
> > > We would like to emphasize that the main contributions of the paper is to provide an explanation of the recent success of simple transfer learning baselines with few-shot learning tasks (Tian et al. 2020, Dhillon et al. 2020). As mentioned in the "Contributions" paragraph (in Section 1), we "... present a new perspective on transfer learning with foundation models, based on the recently discovered phenomenon of neural collapse..." and "... our results provide a compelling explanation for the empirical success of foundation models ...". This is also emphasized in the abstract.
> > >
> > > In fact, prior to our work it was unclear why foundation models can yield better performance in few-shot learning than methods that are specifically designed for few-shot learning, as found by Tian et al. (2020) and Dhillon et al. (2020). We provide an explanation to this intriguing, non-intuitive result, considering one of the typical settings of training foundation models for transfer learning (i.e., one multi-class classification source task and one multi-class classification target tasks), which lacks theoretical understanding in the current state of the literature.
> > > For this purpose, we establish a connection between transfer learning and neural collapse, which is a new perspective on transfer learning.
> > >
> > > In the course of developing our explanation, we use standard tools from learning theory, and we certainly do not claim that the techniques to prove Propositions 1 and 2 (and 5, etc) are novel (this is why we call them propositions and not theorems). We also agree that Proposition 1 is a very much expected result. On the other hand, while Proposition 2 does not have any technical novelty in the proof, we believe it is an important result, as it was non-trivial for us that the generalization of the neural collapse phenomenon to new classes is a key to understand the few-shot performance. Hence, while the techniques used to derive our theoretical results are not novel, the conclusions are new and insightful.

---

### Official Review · Reviewer_BPMK · 2021-11-03

**Correctness:** 4
**Technical Novelty And Significance:** 3
**Empirical Novelty And Significance:** 3
**Recommendation:** 6
**Confidence:** 3

**Main Review:**

I don't have serious concerns but would appreciate it if the authors could expand on the following comments and questions.

- Do you always pick the last layer for features in the foundation model? Did you try selecting second last or third last layers for example? I realize that those layers might not experience neural collapse, but it might be interesting to see how much performance drops. This can also be a baseline to examine whether the neural collapse (using the last layer) is really the reason behind transfer learning performance gains.
- This phenomenon of decreasing variance of the features of a given class (or the concentration of features to mean feature value) seems equivalent to finding invariant features amongst classes. This phenomenon could also be the possible explanation behind the generalization and robustness of the models as explored in IRM [1] (here you can consider each class as a distribution i.e. $P_{c}$). What are your thoughts on this? do you think that there could be an alternate explanation?
- Can the representations be used from a model trained with self-supervised objective instead of cross-entropy loss using labels? or the assumption of finding the foundation model $f$ based on $l$-class classification is very strict (relying on neural collapse phenomenon)?
- I haven't thoroughly checked the proof of proposition 1, but you mention that you expect the generalization gap terms to typically scale as $O(1/\sqrt(m_c))$. Since this is one of the main assumptions could you please point out the circumstances where this scaling might not happen?
- In Section 5.1, while transferring to the few-shot classification task, in addition to training the new linear layer $g$ do you also finetune $f$?
- In Fig.1 the legend (or at least the caption) should indicate that the plots correspond to the number of source classes.

[1] https://arxiv.org/abs/1907.02893

**Summary Of The Paper:**

In this work, the authors investigate the success of transfer learning with foundation models from the perspective of the neural collapse phenomenon. More specifically they study the transfer learning capability of foundation models for few-shot downstream tasks. The paper provides both theoretical and empirical justifications to elaborate on this point.

Note: The paper is heavily inspired by the pre-existing literature. It takes the theoretical perspective from Papyan et al. and somewhat uses that to provide justifications (understanding) of empirical results in Tian et al. and Dhillon et al.

**Summary Of The Review:**

Even though the paper takes heavy inspiration from the previous works, I believe the results are important and would be beneficial for the community. It provides new insights for effective transfer learning in the few-shot setting.

---

> ### Author Response · Authors · 2021-11-23
> **Response to reviewer BPMK**
>
> > Reviewer: Do you always pick the last layer for features in the foundation model? Did you try selecting lower layers?
>
> Answer: We always picked the penultimate layer (top embedding layer)  in the neural network. During the rebuttal we also examined this interesting question to some extent, considering the second-to-last embedding layer as the feature layer. The conducted experiments, presented in Appendix B.2, show that the same neural collapse phenomenon is also present in this layer (together with reasonable few-shot performance). We could not run the experiment on lower layers due to their large dimensions (~30K), leading to memory overflow when applying linear regressions on top of them. Studying the behavior of lower layers is an interesting future work.
>
> > Reviewer: What is the relationship between neural collapse and IRM?
>
> Answer: As far as we understand, IRM tries to solve a somewhat different problem, finding a feature map and a classifier on top of it which works well on multiple environments. Our setting is somewhat different, with even the loss function changing in every problem. However, with a slight reformulation, we can have a fixed loss function (the zero-one loss) and a changing output alphabet (containing only the labels interesting in the actual problem). Using this formulation, one could hope that using the neural collapse of the features, for a large enough network, we can learn a fixed score function for each class (e.g., a fixed function computing the logit value for each possible class), such that the decision in every problem is computed as the maximum score over the classes associated with the given problem. This could work as long as the source classes are representative enough for the target classes (obviously a new score function should be learned for every unseen target class), and the target classes are not significantly denser (in the feature space) than the source classes. In this sense, neural collapse might be a useful tool for analyzing IRM, but this would require substantial work. On the other hand, IRM seems to be well-fitted to the standard transfer-learning setting, while in our few-shot analysis a distinguishing feature was that the loss functions differ (are defined over different sets) in the source and the target tasks.
>
> > Reviewer: Can the representations be used from a model trained with self-supervised objective instead of classification?
>
> Answer: Practically, the representation can be obtained using any methods, including self-supervised learning. Our analysis of few-shot learning depends on the neural collapse phenomenon, and hence applies to any learning method where it is present. We are not aware of any work connecting self-supervised methods with neural collapse, but it also seems an interesting avenue of future work.
>
> > Reviewer: Scaling of the generalization gaps.
>
> Answer: The $O(1/\sqrt{m_c})$ scaling we mentioned is typical for “learnable” function classes (see, e.g., [1,2]). In the revised version we added Proposition 3 (Appendix E) showing that this is indeed the case for ReLU neural networks, and we also provide now a formal analysis in Proposition 4 (Appendix E) for the Rademacher complexity $\mathbb{E}[R(H_{\mathcal{F}^*}(\tilde{\mathcal{P}}))]$ from Proposition 2. The generalization gap becomes larger as the complexity of the function class grows, for example the Rademacher complexity of the class of all mappings $f:\mathbb{R}^d\to [-1,1]^p$ is always 1, and hence the worst-case generalization gap is at least a (positive) constant.
>
> [1] V. Vapnik. “Statistical Learning Theory”. 1989.
> [2] S. Shalev-Schwartz and S. Ben-David. “Understanding Machine Learning: From Theory to Algorithms”. 2014.
>
> > Reviewer: Do you also finetune $f$ during the adaptation to a new task?
>
> Answer: We did not apply fine tuning of the feature map on the target tasks (we made this explicit in Section 5.1 of the revised paper). While fine tuning is obviously a good method to improve performance in few-shot/transfer learning tasks, we wanted to investigate how a trained network works out of the box, and analyze the effects of neural collapse. Also note that in case the network is used for multiple new problems, fine tuning the feature map has the potential problem of degrading the performance on one task while improving on another, keeping the feature map constant does not have this issue.
>
> > Reviewer: Indication that the plots correspond to the number of source classes.
>
> Answer: These are now explicitly stated in the captions.

---

### Decision · Program_Chairs · 2022-01-20

**Decision:**

Accept (Poster)

**Comment:**

Based on the previously observed neural collapse phenomenon that the features learned by over-parameterized classification networks show an interesting clustering property, this paper provides an explanation for this behavior by studying the transfer learning capability of foundation models for few-shot downstream tasks. Both theoretical and empirical justifications are presented to elaborate that neural collapse generalizes to new samples from the training classes, and to new classes as well.

The problem that this paper delves into is important. The paper is well-motivated, and well structured with a good flow. Both theoretical and empirical analyses of the paper are solid. Preliminary ratings are mixed, but during rebuttal, multi-round responses and in-depth discussions were carried out between authors and reviewers, and the final scores are all positive with major concerns well addressed. AC considers the paper itself and all relevant threads, and recommends the paper for acceptance. Authors shall incorporate all response materials into the future version.